# DEL: Discrete Element Learner for Learning 3D Particle Dynamics with Neural Rendering

**Jiaxu Wang[1]**    **Jingkai Sun[1,2]**    **Junhao He[1]**    **Ziyi Zhang[1]**
**Qiang Zhang[1,2]**    **Mingyuan Sun[3]**    **Renjing Xu[1]**
[1] Hong Kong University of Science and Technology, Guangzhou, China
[2] Beijing Innovation Center of Humanoid Robotics Co. Ltd, Beijing, China
[3] Northeastern University, Shenyang, China
{jwang457, qzhang749, jsun444}@connect.hkust-gz.edu.cn
mingyuansun@stumail.neu.edu.cn
{junhaohe, ziyizhang, renjingxu}@.hkust-gz.edu.cn

## Abstract

Learning-based simulators show great potential for simulating particle dynamics when 3D groundtruth is available, but per-particle correspondences are not always accessible. The development of neural rendering presents a new solution to this field to learn 3D dynamics from 2D images by inverse rendering. However, existing approaches still suffer from ill-posed natures resulting from the 2D to 3D uncertainty, for example, specific 2D images can correspond with various 3D particle distributions. To mitigate such uncertainty, we consider a conventional, mechanically interpretable framework as the physical priors and extend it to a learning-based version. In brief, we incorporate the learnable graph kernels into the classic Discrete Element Analysis (DEA) framework to implement a novel mechanics-integrated learning system. In this case, the graph network kernels are only used for approximating some specific mechanical operators in the DEA framework rather than the whole dynamics mapping. By integrating the strong physics priors, our methods can effectively learn the dynamics of various materials from the partial 2D observations in a unified manner. Experiments show that our approach outperforms other learned simulators by a large margin in this context and is robust to different renderers, fewer training samples, and fewer camera views.

## 1 Introduction

Simulating complex physical dynamics and interactions of different materials is crucial in areas including graphics, robotics, and mechanical analysis. While conventional numerical tools offer plausible predictions, they are computationally expensive and need extra user inputs like material specifications. In contrast, learning-based simulators have recently garnered significant attention as they offer more efficient solutions. Previous works primarily simulate object dynamics in 2D. They either treat pixels as grids [1] or map the images into low latent space [2, 3] and predict future latent states. However, these 2D-centric approaches possess limitations. The world is inherently 3D, and 2D methods struggle to reason about physical processes because they rely on view-dependent features, and are hard to understand real object geometries. To address these, researchers have incorporated multi-view 3D perceptions into simulations such as Neural Radiance Field (NeRF) [4] which is an implicit 3D-aware representation. Some studies extract view-invariant representations and 3D structured priors by NeRFs to learn 3D-aware dynamics [5, 6]. They either represent the whole scene as a single vector or learn compositional object features by foreground masks. However, they require heavy computational demands and struggle with objects with high degrees of freedom.

38th Conference on Neural Information Processing Systems (NeurIPS 2024).

Particle-based learned simulators show impressive results in modeling 3D dynamics. The success is mainly attributed to the popularity of Graph Neural Networks (GNNs). In general, objects are represented as particles that are regarded as nodes in graphs, and their interactions are modeled by edges. Previous studies [7, 8] adapt GNN to predict particle tracks and achieve good results. Recent research has made strides in improving GNN simulators [9, 10]. They require particle correspondences across times for training. However, 3D positions of particles across time are not always accessible and learning dynamics solely from visual input is still a big challenge. The reasons can be summarized as follows. Determining particle positions from 2D observations leads to uncertainty since different particle distributions can produce similar 2D images. Moreover, previous GNN-based simulators aim to learn how to infer the entire dynamics, which are fully uninterpretable and result in hard optimization. Several studies [11, 12] reconstruct 3D from 2D images and then learn from it, but they are not directly trained end-to-end with pixel supervision. One feasible way is to employ inverse rendering. For example, [13, 14] use NeRF-based inverse rendering to learn dynamics from 2D images. However, they are either constrained to simulate specific material or incapable of dealing with the 2D-3D ambiguities, thereby damaging their generalization ability. Furthermore, existing approaches only evaluate their methods on simple datasets. Their synthetic dataset often contained a limited variety of materials (usually rigid bodies), rarely involved collisions between objects, and featured very regular initial shapes.

To address the above challenges, this work incorporates strong mechanical constraints in the learning-based simulation system to effectively learn the 3D particle dynamics from 2D observations. Discrete Element Analysis (DEA) [15], also known as Discrete Element Method [16], is a traditional numerical method to simulate particle dynamics in mechanical analysis. This method computes interaction forces between particles to predict how the entire assembly of particles behaves over time. However, traditional DEA heavily relies on the user-predefined mechanical relations between particles, such as constitutive mapping or dissipation modeling, which often involve several material-specific hyperparameters. Moreover, the results often deviate from the actual mechanical responses because the constitutive equations are overly idealized.

In this work, we combine GNNs with the DEA theory to implement a physics-integrated neural simulator, called the Discrete Element Learner (DEL), aiming to enhance the robustness, generalization, and interoperability of GNN architectures. In detail, we use GNNs as kernels to learn the mechanical operators in DEA rather than adopting user-define equations. On the other hand, the graph networks only need to fit some specific physical equations in the DEA framework instead of learning the whole evolution of dynamics, which largely reduces the optimization difficulties. Therefore, combining the conventional mechanical framework and GNN can reduce the 2D-to-3D uncertainty and alleviate the ill-posed nature. The main contributions of this work can be summarized as follows:

- We propose a novel physics-integrated neural simulation system called DEL which incorporates graph networks into the conventional Discrete Element Analysis framework to effectively learn the 3D particle dynamics from 2D observations in a physically constrained and interpretable manner.

- We design the network architecture under the guidance of the DEA theory. In detail, we use learnable GNN kernels to only fit several specific mechanical operators in the classic DEA framework, instead of learning the entire dynamics to make GNNs and the DEA mutually benefit from each other, significantly reducing the ill-posed nature of this task.

- We evaluate our method on synthetic datasets that contains various materials and complex initial shapes compared to existing ones. Extensive experiments show that our method surpasses all previous ones in terms of robustness and performance.

## 2 Related Work

### 2.1 GNN-based particle dynamics simulator

There has been many works [17, 18, 9, 19, 20, 21] to develop GNN-based particle simulators to predict 3D dynamical systems. This is because representing 3D scenes as particles perfectly matches the graph structure via particles as nodes and interactions as edges. GNS [7] shows plausible simulations on multiple materials by multi-step message passing. DPI [8] adds one level of hierarchy to the rigid and predicts the rigid transformation via generalized coordinates. EGNN [17] maintains the

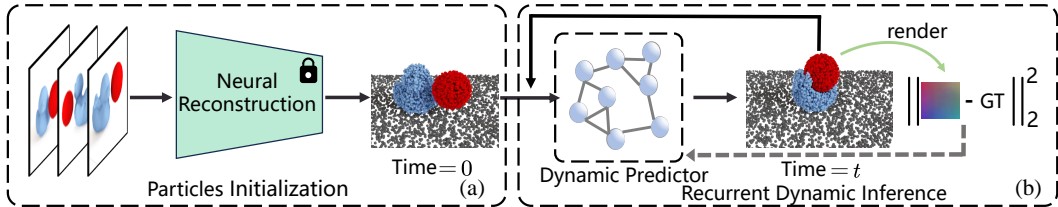

Figure 1: The paradigm of the dynamics learning via inverse rendering. (a) Particles Initialization Process. The scene is initialized as particles. (b)Recurrent Dynamic Inference Process. The generated particle set is fed into a dynamic predictor to infer the next state iteratively.

equivariance of graphs by passing scalar and vector messages separately, and explicitly assumes the direction of vector message passing along with the edges. SGNN [9] proposes the subequivariant simulator, which has a strong generalization to long-term predictions. Most of them are black-box models and non-interpretable, thus complicating the optimization. There are some works incorporating basic physical priors into neural networks [22, 23, 24, 25, 26], whereas they perform well either on toy examples or specific topologies such as rigid hinges. All the above-mentioned learned simulators require full 3D particle tracks as labels for training. They cannot learn from pixel-level supervision because of the large solution spaces caused by the 2D-to-3D uncertainties, which we experimentally proved in Section 4. Our approach reduces ambiguities by integrating GNNs into a mechanical analysis framework. Our method not only yields impressive results supervised by 3D labels but also effectively learns realistic physics under 2D supervision.

## 2.2 Learning dynamics from 2D images

Learning dynamics from merely visual observations is vital for many domains. Previous works [1, 3] map images into low dimensional space and learn dynamic models to infer the evolution of latent vectors. However, the general latent approach [27, 28, 2, 29, 30, 31, 32, 33] makes things like pixel-level video prediction rather than real physical inference [34]. The biggest reason is the gap between 2D observations and 3D worlds [10]. To address this challenge, recent works consider 3D-invariant representation to build latent states. NeRF is used by [5] to encode multiview images as view-independent features. But it serves the whole scene as a single vector, and cannot handle scenes with multiple objects. [6] encodes compositional multi-object environments into implicit neural scatter functions, while it only handles rigid objects. Similarly, [33] and [35] use compositional implicit representation, but cannot simulate objects with large deformations. Some other methods [36, 37, 38] need additional signals, such as Lidar data. The 3D-aware latent dynamics also lack generalizability to unobserved scenarios and cannot work with complex topologies and varying materials. Moreover, latent dynamics models are fully non-interpretable.

Very recently, with the development of differentiable neural rendering, a few studies have attempted to train 3D dynamic models from visions via inverse rendering [13, 14, 39]. They bridge images and 3D scenes with a differentiable renderer to minimize the renderings and groundtruth. However, [13] and [14] can only simulate fluids because they require fluid properties as input. VPD[39] learns 3D particle dynamics directly from images and can simulate various solid materials. However, it requires jointly training its own particle renderer and latent simulator, which leads to a black-box nature and is hard to be adopted by other renderers. Conversely, our DEL can seamlessly integrate into any point-based renderers with satisfactory performance and is physically interpretable as well as can simulate various materials in a unified manner.

## 3 Methodology

### 3.1 Preliminaries and Problem Statement

This task is formulated as inferring particle dynamics via inverse neural rendering. Similar to other inverse graphics, the scene can be represented by 3D primitives, and then the dynamical module infers the future state of these primitives. Once this future state is inferred, it can be effectively transformed into visual images by neural renderers. The dynamical module can be trained from

the error between the renderings and observations. Figure 1 illustrates the general paradigm. In this formula, 3D scenes should be represented as particles, and then, the renderer should be able to render particles into images with a given camera viewpoint. According to the above discussion, we choose the Generalizable Point Field [40] (GPF) as our renderer, which can represent a 3D scene as particles, change its content by moving particles, and render images with arbitrary views. Notably, other particle-based renderers such as [41, 13] or recently prevailing [42], can also be used arbitrarily as long as they are fully differentiable and represent objects as particles. The renderer module can iteratively produce updated images after the dynamic module moves particles at each timestamp.

The dynamical module operates as a graph network simulator. Consider a physical system with N particles to represent M objects, the simulator models its dynamics by mapping the current state to consequent future states, usually the positions of particles. Assume $X_i^t \in X$ are particle states at time t, $X_i^t$ usually includes the coordinate $\mathbf{x}_i^t$, the velocity $\mathbf{v}_i^t$ and particle's intrinsic attributes $\mathbf{A}_i$ such as the material type and mass. The learnable GNN simulator $S$ considers particles as nodes and dynamically constructs connections at independent time steps when the distance between two particles is smaller than a threshold ($E = \{i, j : ||\mathbf{x}_i - \mathbf{x}_j||_2 <= r\}$). The GNN maps all the information at the current state to the positions at the next timestamp by passing messages on the graph, i.e. $\mathbf{x}_i^{t+1} = S_\theta(\mathbf{x}_i^t, \mathbf{v}_i^t, \mathbf{a}_i, E)$. Different GNN simulators mainly lie in the different designs of message-passing networks. As we claim in Section 1, the $S_\theta$s in most previous approaches aim to learn the entire dynamics process, which leads to hard optimization and the risk of overfitting. Moreover, they are non-interpretable black boxes. Therefore, the learning target would be very ill-posed because the solution space is very large when only visual supervisions are given.

We propose a mechanics-encoded architecture that combines the GNN with the typical DEA to reduce uncertainty and improve interpretability. In the following section, we first introduce the general DEA method and its potential to be enhanced by the learning-based kernels. Second, we present how we incorporate the graph networks into DEA to replace the traditional operators.

## 3.2 The General Discrete Element Analysis Theory

In this section, we introduce the general framework of Discrete Element Analysis, also known as the Discrete Element Method, and its drawbacks which potentially can be enhanced by our learnable kernels. Here we only cover the general knowledge that we need to design our architecture, we recommend readers refer to [15, 43, 16] for deeper knowledge of DEA. In the framework, the whole scene is represented as particles and the DEA is used to simulate the behavior and interactions of these particles. Generally, in this framework, the movement of an individual particle is governed by the Newton-Euler motion equation:

$$m_i \frac{d^2\mathbf{u}}{dt^2} = \sum_{j=1}^{n}(\mathbf{F}_{ij}^p + \mathbf{F}_{ij}^v) + \mathbf{F}_i^g \tag{1}$$

where $\mathbf{u}$ is the movement vector, $m_i$ is the mass of particle $i$. $F_i^g$ refers to the gravity. $F_{ij}^p$ and $F_{ij}^v$ are the interaction forces between particle $i$ and $j$, the former marks potential interaction force, and the latter marks dissipative (viscous) contributions. The potential interactions primarily arise from physical contact between elements [15]. The dissipative contributions take into account kinetic energy dissipation mechanisms concerned with the dispersion of elastic waves (this dissipation is general for all materials) [43]. Given this context, the potential contributions to interactions assume a significant role while the dissipative contribution merely influences the energy dissipation within the system. Therefore, the fundamental problem is to formulate a general form of potential interactions between particles, which would apply to materials with different features of mechanical responses.

Besides, $\mathbf{F}_{ij}^p$ and $\mathbf{F}_{ij}^v$ can be decomposed into the normal and tangential directions, which are represented by the superscript $n$ and $t$ in Equation 2.

$$\mathbf{F}_{ij}^p = \mathbf{F}_{ij}^{pn} + \mathbf{F}_{ij}^{pt}, \ \mathbf{F}_{ij}^v = \mathbf{F}_{ij}^{vn} + \mathbf{F}_{ij}^{vt} \tag{2}$$

Substituting Equation 2 into Equation 1 and omitting the gravity terms for simplicity, we can derive:

$$m_i \frac{d^2\mathbf{u}}{dt^2} = \sum_{j=1}^{N}(\mathbf{F}_{ij}^{pn} + \mathbf{F}_{ij}^{vn}) + \sum_{j=1}^{N}(\mathbf{F}_{ij}^{pt} + \mathbf{F}_{ij}^{vt}) \tag{3}$$

where the first term is the normal constituent and the second term is the tangential constituent. Here we discuss the potential interaction forces within the two directions respectively. According to [16], The $\mathbf{F}_{ij}^{pn}$ can be considered as the composition of contact force $f_{ij}^{cn}$ and bond force $f_{ij}^{bn}$. The contact forces are activated when two particles physically contact and collide. The bond forces mean if the two particles belong to the same object, they are connected by a bond that will provide attraction or repulsion based on their relative positions to maintain the fundamental properties of the material. We depict the mechanism in Figure 2. Furthermore, in DEA, a physical quantity called intrusion scalar is commonly used to compute the two forces:

$$\delta d_n = (r_i + r_j) - \|\mathbf{x_i} - \mathbf{x_j}\|_2 \tag{4}$$

where $r_i, r_j$ are the radius of particle $\mathbf{x_i}, \mathbf{x_j}$. $\delta d_n > 0$ means they contact and vice versa. We use visual aid Fig. 2(a) to depict $\delta d_n$ intuitively. The red hard sphere intrudes into the blue soft sphere in the figure. The deformation length on the blue surface refers to the $\delta d_n$. The normal contact force $f_{ij}^{cn}$ should be related to the $\delta d_n$ because the particle tends to recover its initial shape. In addition, the direction of $f_{ij}^{cn}$ is the normal direction between $i$ and $j$. On the other side, as stated in Figure 2(b), $f_{ij}^{bn}$ acts as a linkage between two particles belonging to the same object, akin to a bond. $f_{ij}^{bn}$ also relates to $\delta d_n$. According to the above discussion, the total normal potential interaction forces can be formulated as:

$$\mathbf{f}_{ij}^n = \begin{cases} (\mathcal{F}_c^n(\delta d_n, A_{ij}) + \mathcal{F}_b^n(\delta d_n, A_{ij}))\mathbf{n} & i, j \in \mathcal{O}_k, \\ \mathcal{F}_c^n(\delta d_n, A_{ij})\mathbf{n} & i, j \notin \mathcal{O}_k. \end{cases} \tag{5}$$

where $\mathcal{F}_c^n, \mathcal{F}_b^n$ are two functions to map the $\delta d_n$ and $A_{ij} = [A_i, A_j]$ to the contact and bond forces respectively. $\mathcal{O}_k$ is the Object k. Here $A_*$ is the material properties in the particles' small surrounding vicinity. $\mathbf{n} = \frac{\mathbf{x_j} - \mathbf{x_i}}{\|\mathbf{x_j} - \mathbf{x_i}\|_2}$ is the normal unit vector. In DEA, the $\mathcal{F}_c^n, \mathcal{F}_b^n, A_i, A_j$ usually require to be specified by users, which are often simple linear or polynomial functions. For example, the most simple way to compute the total interaction force is the linear spring model $\mathcal{F}_{c+b}^n = k\dot{\delta}d_n$. Some more complex functions such as Hertz-Mindlin (Eq. 6) are also commonly used.

$$F_c^n(\delta d_n) = \frac{4}{3}E\sqrt{R}\delta d_n, F_b^n(\delta d_n) = k_b^n \delta d_n \tag{6}$$

In Eq. 6, $E, R, k_b$ are human-defined material parameters. However, the above-introduced handcrafted mapping functions only roughly approximate real cases and always deviate from the realistic natures of materials, leading to inaccurate simulations of DEA. Likewise, the tangential interactions can be analogous to the above. Notably, the normal direction contributes mostly to the total potential interactions because the tangential deformation is small due to the friction constraints [44]. Therefore, general DEA often approximates it by the instantaneous displacement within a timestamp $\Delta t$, i.e. $\delta d_t = \|\mathbf{v}_{ij}^t \Delta t\|_2$. $\mathbf{v}_{ij}^t$ is the tangential velocity of $j$ relative to $i$.

$$\mathbf{f}_{ij}^{t'} = \mathcal{F}_c^t(\delta d_t, A_{ij}, \|\mathbf{f}_{ij}^n\|)\mathbf{t} + \mathcal{F}_b^n(\delta d_t, A_{ij}, \|\mathbf{f}_{ij}^n\|)\mathbf{t} \tag{7}$$

Similar to $\mathbf{n}$, the $\mathbf{t} = \mathbf{v}_{ij}^t/\|\mathbf{v}_{ij}^t\|$ is the tangential unit vector to determine the force direction. $\mathcal{F}_c^t, \mathcal{F}_b^t$ are the user-defined simple functions. Furthermore, the tangential magnitude is also affected by the normal force [45] thus $\|\mathbf{v}_{ij}^t\|$ is included. The DEA theory includes the dissipative contribution [15], also called the viscosity term [45], as it is only to simulate energy dissipation to prevent the system from exhibiting perpetual motion. In general, large velocities lead to large dissipation. In the general DEA, the term is often modeled by:

$$F_{vis} = -\eta \cdot c_{crit} \cdot v_{rel} \tag{8}$$

where $c_{crit}$ is a material properties called critical damping, $v_{rel}$ refers to $v_n$ (normal velocity), $v_t$ (tangential velocity) correspondingly.

Based on the above discussion, DEA requires multiple user-specific mechanical operators including $F_c^n, F_b^n, F_{vis}^n, F_c^t, F_b^t, F_{vis}^t$, which would be inaccurate and fully different for various materials. Moreover, we cannot precisely estimate the real properties of materials when only videos are available. This can be considered another impracticability of the DEA framework. On the contrary, the benefit of DEA is that we only need to solve the magnitudes of these decomposed forces because these directions are physically constrained in this framework. Therefore, we keep the advantages of the physical priors while remedying the defects of DEA by replacing these mechanical operators with trainable network kernels.

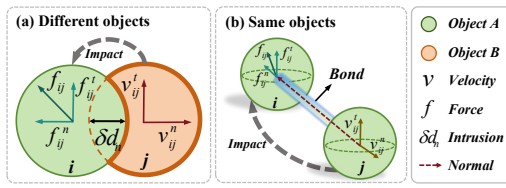

Figure 2: Two cases of particle interactions. (a) contact forces, affected by the intrusion $\delta d_n$. (b) The bond force exists between two particles of the same object, affected by the bond length.

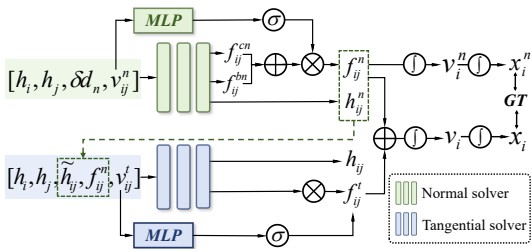

Figure 3: The main pipeline of message-passing network

## 3.3 Mechanics-informed Graph Network Architecture

This subsection introduces the proposed Discrete Element Learner which replaces these human-specific operators in DEA with learnable graph kernels. In other words, we integrate physics prior knowledge into the network design to make the entire AI system differentiable and can be optimized through image sequences. We visualize such mechanics-integrated network architecture in Fig. 3. Furthermore, our method implicitly encodes the material properties into embedding vectors during the unsupervised training. Similar to other GNN-based simulators, we first construct subgraphs by searching neighbors with a fixed radius, and each subgraph can be viewed as a collective of particles involved in interactions. Then we convert all physical variables including velocities and positions to each particle-centric coordinate system when analyzing associated particles.

We define the DEA-incorporated message-passing network as follows and the symbols previously used maintain their consistent meanings. First, we encode each particle attribute $A_i$ such as material types into latent embedding $h_i \in R^{200}, [-1, 1]$ via Eq. 9.

$$h_i = Norm(\text{MLP}(A_i)) \tag{9}$$

Second, the following four equations in GNN are used to implement Eq. 5

$$n_i, n_j, e_{ij} = \Phi^n(\delta d_n, h_i, h_j) \tag{10}$$

$$f_{ij}^{cn} = \text{ReLU}(\mathcal{H}_c(n_i, n_j, e_{ij})) \tag{11}$$

$$f_{ij}^{bn} = \mathcal{H}_b(n_i, n_j, e_{ij}) \tag{12}$$

$$f_{ij}^{n'} = f_{ij}^{cn} + f_{ij}^{bn} \tag{13}$$

where $\Phi^n$ is a graph network kernel, $n_i, n_j$, and $e_{ij}$ are node and edge features. $n_{i,j}$ encode the properties of the small regions around particle $i, j$. $e_{ij}$ encodes their interaction. $\mathcal{H}_c$, and $\mathcal{H}_b$ are two heads to regress the magnitude of the two forces. Due to the previous discussion, the contact force $f_{ij}^{cn}$ can only act from $j$ towards $i$, it must be a positive value, therefore we apply ReLU activation. While the bond force $f_{ij}^{nb}$ can be either positive or negative, thereby no activation is used. As for the dissipative effect, we consider the dissipation as a reduction coefficient rather than directly regress its value because it always diminishes the potential interaction force. In our network, we use an MLP $\phi^n$ activated by Sigmoid $\sigma$ to model the normal dissipative phenomenon in Eq. 14.

$$\mathbf{f}_{ij}^n = \sigma(\phi^n(\|v_{ij}^n\|_2, e_{ij})) f_{ij}^{n'} \mathbf{n} \tag{14}$$

Likewise, we apply another kernel $\Phi^t$ (Eq. 15) to replace Eq. 7. A minor difference is that we omit the tangential bond force because when the particle undergoes very small relative displacement tangentially, the bond length remains nearly unchanged ($\mathbf{f}_{ij}^{bt} \approx 0$). In this way, $\mathbf{f}_{ij}^t = \mathbf{f}_{ij}^{ct}$.

$$f_{ij}^{t'}, e_{ij} = \Phi^t(\|v_{ij}^t\|_2, f_{ij}^{n'}, e_{ij}, n_i, n_j) \tag{15}$$

According to the discussion in the previous section, the quantity $f_{ij}^{t'}$ relates to particle properties ($n_*$), the tangential relative displacement ($\|v_{ij}^t\|_2 \Delta t$), and the precomputed normal pressure ($f_{ij}^{n'}$). Similar to $\phi^n$, $\phi^t$ models tangential dissipation in Eq. 16.

$$\mathbf{f}_{ij}^t = \sigma(\phi^t(\|v_{ij}^t\|_2, e_{ij})) f_{ij}^{t'} \mathbf{t} \tag{16}$$

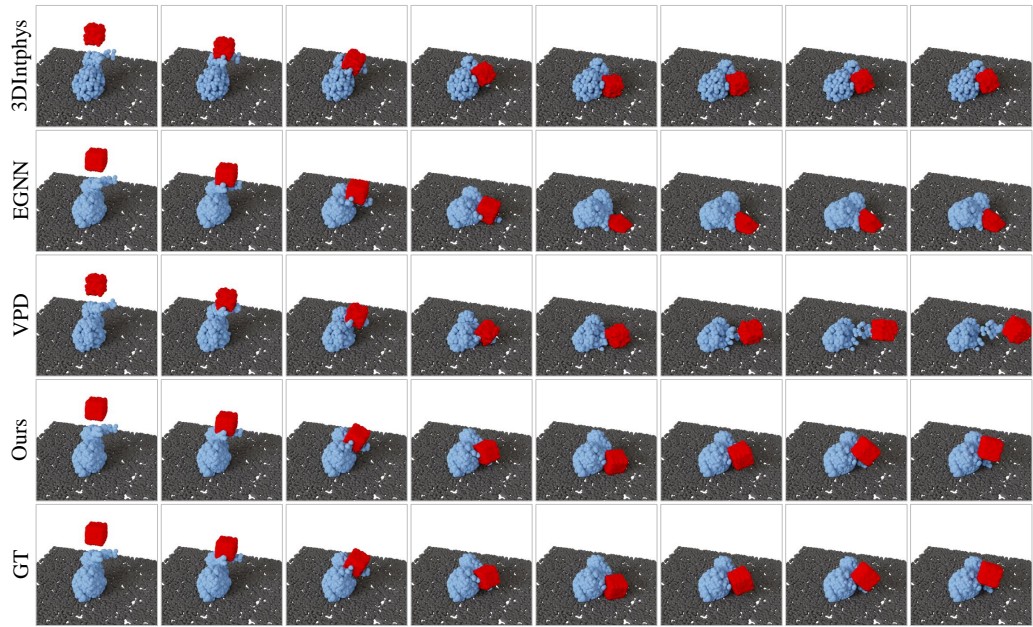

Figure 4: Qualitative Comparisons of dynamics prediction between our DEL and baselines in the particle-view on test sequences.

Notably, all graph kernels output the magnitudes of these mechanical vectors because our mechanical framework precomputes their directions by $\mathbf{n}$ and $\mathbf{t}$, which largely reduces the ambiguities when learning dynamics. It can be observed that our approach deeply integrates the mechanical analysis framework, and is partially interpretable. The design of this architecture is strongly inspired by the physical knowledge to reduce the learning burden, we vividly demonstrate it in Fig. 3.

### 3.4 Training Strategy

We train the DEL from only visions. Assuming cameras record many dynamic sequences including the same materials but with different initial conditions, we first initialize scenes as particles by GPF at timestamp $t_0$. Then we adopt the first three frames to approximate the initial velocities, similar to [46]. Next, the dynamic module tries to move particles to the next state. Then the GPF renders the after-moving particles into images. If the positions are correctly predicted, the rendered images should align with the recordings. We use L2 loss to supervise

$$\mathcal{L}_r = \sum_{cam} \|I_t^{cam} - R(D(x_{t-1}), cam)\|_1 \tag{17}$$

where $D$ refers to the dynamic module, $X_{t-1}$ is particle states at the last timestamp, $R$ is the GPF rendering module, and $I_t^{cam}$ is the observed image at view $cam$ and time $t$. Moreover, we apply the same L2 loss to supervise the images produced by the particles updated by using only the $\mathbf{f}_i^n$ (the normal force) as stated in Figure 3. This aims to amplify the contribution of the normal direction since it is claimed in the previous section that the normal direction is the main influencing factor for collisions. The formulation of this L2 loss is referred to as $\mathcal{L}_r^n$. In addition, the gradients of rendering loss $\frac{\partial \mathcal{L}_r}{\partial \mathbf{x}}$ can be interpreted as the scaled velocities of particles. Thus we propose the gradient loss to constrain the direction of the total velocity of each particle.

$$\mathcal{L}_g = \sum_{i=1}^N (\frac{\partial L_r}{\partial x_i} / \left\|\frac{\partial L_r}{\partial x_i}\right\|_2 - \frac{v_i(t)}{\|v_i(t)\|_2}) \tag{18}$$

The final loss is $\mathcal{L} = \mathcal{L}_r^n + \mathcal{L}_r + \beta\mathcal{L}_g$. More implementation details are contained in the Appendix.

## 4 Experiments

**Dataset.** Existing datasets in this field contain only a few types of materials or even only rigidity, lack rich interactions between objects, and have simple initial shapes. Therefore, we create a more

Table 1: Quantitative Comparisons between ours and benchmarks on five scenarios in render views.

| | Plasticine | | | SandFall | | | Multi-Objs | | | FLuidR | | | Bear | | |
|---|---|---|---|---|---|---|---|---|---|---|---|---|---|---|---|
| Method | PSNR↑ | SSIM↑ | LPIPS↓ | PSNR↑ | SSIM↑ | LPIPS↓ | PSNR↑ | SSIM↑ | LPIPS↓ | PSNR↑ | SSIM↑ | LPIPS↓ | PSNR↑ | SSIM↑ | LPIPS↓ |
| SGNN* [9] | 25.27 | 0.925 | 0.143 | 23.61 | 0.886 | 0.216 | 24.76 | 0.909 | 0.166 | 28.88 | 0.935 | 0.168 | 27.61 | 0.949 | 0.132 |
| NeRF-dy [5] | 21.09 | 0.893 | 0.225 | 22.58 | 0.879 | 0.216 | 19.61 | 0.826 | 0.318 | 25.79 | 0.925 | 0.270 | 22.83 | 0.873 | 0.232 |
| EGNN* [17] | 26.27 | 0.944 | 0.119 | 25.17 | 0.918 | 0.178 | 26.38 | 0.928 | 0.144 | 30.28 | **0.951** | 0.123 | 29.13 | 0.953 | 0.117 |
| VPD [39] | 27.06 | 0.941 | 0.101 | 24.61 | 0.926 | 0.127 | 25.62 | 0.921 | 0.136 | 30.06 | 0.947 | 0.126 | **30.52** | **0.964** | **0.102** |
| Ours | **28.09** | **0.959** | **0.091** | **26.65** | **0.945** | **0.113** | **27.06** | **0.939** | **0.128** | **30.53** | 0.944 | **0.122** | 30.08 | **0.964** | 0.105 |

Table 2: Quantitative comparisons between ours and baselines on five scenarios in particle views.

| | Plasticine | | SandFall | | Multi-Objs | | FluidR | | Bear | |
|---|---|---|---|---|---|---|---|---|---|---|
| Method | CD↓ | EMD↓ | CD↓ | EMD↓ | CD↓ | EMD↓ | CD↓ | EMD↓ | CD↓ | EMD↓ |
| SGNN* [9] | 35.91 | 26.4 | 2.47 | 2.69 | 20.3 | 26.9 | 3.98 | 5.02 | 4.69 | 5.01 |
| 3DIntphys [11] | 26.99 | 22.61 | 3.17 | 3.35 | 16.55 | 17.61 | 6.92 | 8.01 | 6.69 | 6.01 |
| EGNN* [17] | 16.20 | 14.61 | 2.13 | 2.56 | 13.21 | 13.77 | 2.58 | 3.01 | 3.95 | 4.16 |
| VPD [39] | 16.96 | 12.77 | 1.99 | 2.35 | 14.26 | 14.57 | 3.22 | 2.94 | **3.41** | 3.71 |
| Ours | **7.54** | **7.10** | **1.73** | **1.90** | **8.48** | **9.13** | **1.72** | **1.88** | 3.54 | **3.33** |

challenging dataset that includes various materials (rigid, elastic, plastic, fluid, and sandy soil). The dataset includes six main scenarios. Each scenario contains 128 training and 12 testing dynamic sequences with different initial conditions such as velocities and shapes. These sequences are generated by our MPM simulator, then Blender is used to render them to produce high-fidelity multiview images. We use 4 cameras to observe each dynamic episode. The six scenarios are: **Plasticine** illustrates the collision of an elastic ball with a plasticine toy. **Multi-Objs** includes rigid and elastic objects. **SandFall** depicts sand descending onto elasticities. **Fluids** contains fluid with different viscosity. **FluidR** describes that Newtonian fluid flows onto rigid bodies. **Bear** involves interactions of elastic, plastic, and rigid objects.

**Baselines**. We compare the proposed approach with the two most recent baselines, i.e. VPD [39] and 3DIntphys [11]. These two methods focus on learning 3D particle dynamics from 2D images. VPD must jointly train its specific point renderer and particle dynamics module, which are highly coupled. Therefore, it cannot use other point-based renderers. In contrast, our DEL does not contain any assumption of the neural renderer and thus can be directly adapted to different renderers. We choose the GPF [40] as our renderer. 3DIntphys first reconstructs a series of NeRFs for the image sequences at all timestamps, then extracts the point cloud from each NeRF, and finally trains its dynamics module with point similarity loss across time. Hence, it cannot be trained in an end-to-end manner and cannot render novel views from the updated particle sets.

Moreover, to evaluate the effectiveness of our dynamics prediction. We set two additional baselines in which we replace our dynamic model with two prevailing GNN-based simulators, i.e. EGNN [17] and SGNN [9] while keeping the same point-based render with us. In their original paper, they require 3D particle correspondence across time for training. But in our setting, we also utilize inverse rendering to train them from images. We mark these two variants via an upright *. We also compare with the NeRF-dy [5], a fully implicit dynamics predictor. Next, we additionally retrain NeuroFluid [13] on the Fluids scenario for comparisons because it only supports inferring fluid dynamics.

**Metrix**. We use the Chamber Distance (CD) and Earth Mover Distance (EMD) to measure the similarities between the predicted particle distributions and the groundtruth because no per-particle correspondence is available. The reported CD and EMD are multiplied by $10^2$ in all tables for clarity. We compare the PSNR, SSIM, and LPIPS (AlexNet) between the renderings and 2D labels. We also provide qualitative results and comparisons for a better visual assessment.

## 4.1 Results and Comparisons

**Results in Rendering View.** We also present the comparisons of rendering qualities to further evaluate the effectiveness because we adopt inverse rendering to bridge 2D and 3D. The quantitative results are listed in Table 1. The VPD overall achieves the second-best performance because it is designed specifically for high-quality rendering. However, due to the robust physical priors, our approach more easily learns the underlying physical rules, resulting in a more accurate dynamics prediction, consequently, rendering more plausible images. The other two graph-based simulators perform mediocrely. We additionally show visual examples in Figure 5. Our approach gives better renderings than baselines due to better dynamic predictions. Due to page constraints, we only give

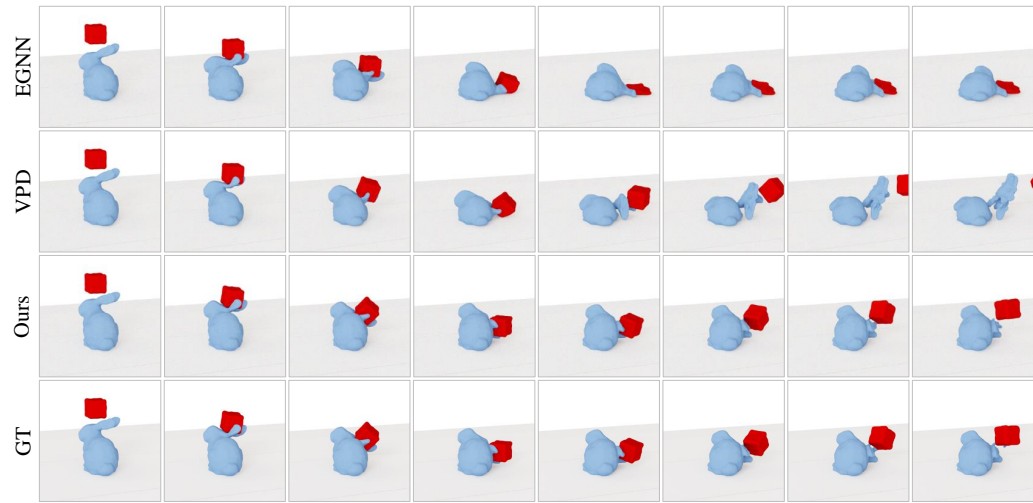

Figure 5: Qualitative Comparisons of rendered images between ours and baselines.

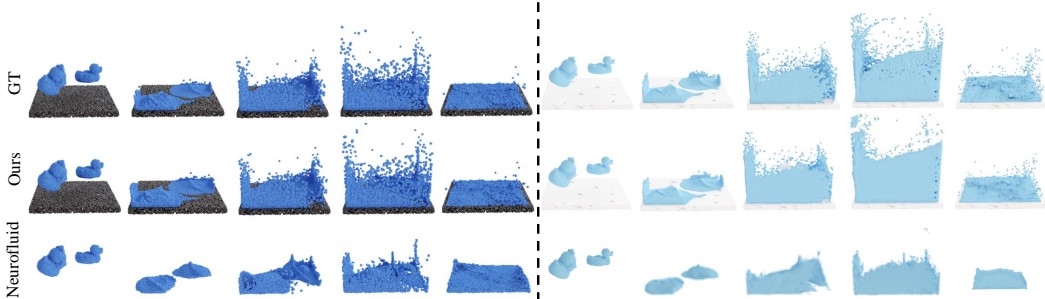

Figure 6: Qualitative comparisons between neurofluid and ours.

examples of the Plasticine scenario. More results on other scenarios can be seen in the **Appendix**.

**Results in Particle View.** In this section, we report the simulation results generated by different approaches in the particle view. Table 2 and Figure 4 show the quantitative and qualitative comparisons respectively. It is observed from Table 2 that our method delivers the most satisfactory results across all scenarios. One interesting finding is that the EGNN* overall outperforms the SGNN*, but in their respective original papers, the SGNN performs better than EGNN when 3D labels are available. The reason might be EGNN benefits from predefining message-passing directions, but SGNN simultaneously determines both directions and values, which is hard to optimize when only 2D images are given. From this figure, VPD, 3DIntphys, EGNN*, and SGNN* cannot predict precise interactions while our method shows steady long-term simulation.

Table 3: Ablation studies of four components

| | SandFall | | Multi-Objs | |
|---|---|---|---|---|
| Method | CD↓ | EMD↓ | CD↓ | EMD↓ |
| $No/\mathcal{L}_g$ | 3.95 | 4.09 | 29.7 | 32.9 |
| $No/\mathbf{f}_{ij}^t$ | 2.26 | 2.78 | 11.4 | 18.6 |
| $No/decomp$ | 3.15 | 3.04 | 16.5 | 13.3 |
| $No/L_n^r$ | 2.65 | 2.91 | 10.27 | 12.38 |
| Full | 1.73 | 1.90 | 8.48 | 9.13 |

Table 4: Quantitative results on Fluids scene

| | Particle-view | | Render-view | | |
|---|---|---|---|---|---|
| Method | CD↓ | EMD↓ | PSNR↑ | SSIM↑ | LPIPS↓ |
| Neurofluid | 10.7 | 11.0 | 25.36 | 0.930 | 0.175 |
| SGNN* | 11.87 | 11.32 | 27.68 | 0.946 | 0.182 |
| EGNN* | 10.76 | 9.78 | 29.01 | **0.962** | 0.108 |
| Ours | **4.18** | **2.97** | **30.02** | 0.962 | **0.104** |

## 4.2 Additional Comparisons and Analysis

**Comparisons with Neurofluid.** Neurofluid [13] is another unsupervised method for learning fluid dynamics. It uses the particle-based PhysNeRF as the renderer and employs DLF [47] as the dynamic modules. We compare our approach with it on the **Fluids**. The results are reported in Table 4 and Fig. 6. The results show that Neurofluid cannot work well on test data because its dynamic module

lacks enough physical priors. Another reason is that it jointly trains the renderer and dynamics modules which makes them compensate for each other, causing overfitting of training data.

**Ablation Studies.** We evaluate some significant components of our method. First, we ablate the gradient loss (marked as $No/\mathcal{L}_g$). Second, we report the contribution of the tangential decomposition constituent $\mathbf{f}_{ij}^t$ ($No/\mathbf{f}_{ij}^t$). Third, we make the graph network fully regress the direction and magnitude of the interaction forces ($No/decomp$) instead of using the priors encoded in the DEA framework, i.e. the output of the graph is a force vector. Next, we ablate the $L_n^r$ loss term, further proving the significance of normal components. The quantitative results are listed in Table 3, which shows that the $\mathcal{L}_g$ and $\mathcal{L}_n^r$ contribute to simplifying the optimization. In addition, the mechanical decomposition is important as well. Even though the main direction of message passing is along the directions of edges, the tangential components indeed make the simulation results more realistic. Furthermore, we evaluate the effect of different renders, different training data sizes, different numbers of cameras used to capture scenes, and different points. We also test the Rollout MSE of the three methods when the 3D labels are available. Both of the results can be seen in our appendix, which shows that our method is satisfactory and robust under all the above ablation conditions.

# 5 Conclusion and Limitation

**Conclusion.** We propose the DEL which combines the Discrete Element Analysis framework with graph networks to effectively learn 3D particle dynamics from only 2D images with various materials. The main idea is to integrate strong physical priors to reduce 2D to 3D uncertainties. Existing GNN-based simulators, which are designed for learning from 3D particle correspondence, try to model the whole dynamics of particles. Differently, the DEL only adopts graph networks as learnable kernels to model some specific mechanical operators in the DEA framework, while keeping its mechanical priors, such as the direction of forces and decompositions of forces. We also evaluate our approach on synthetic data with various materials, initial shapes, and extensive interactions. The experiments show our method outperforms baselines when only 2D supervision is accessible. We also show the robustness of our methods to the renderers, training data sizes, and 3D labels.

**Limitation.** Currently, studies in this field, including this work, are conducted on synthetic datasets due to the impracticability of collecting multiview dynamic videos. Hence, "learning from few data" could potentially help address the problem of learning 3D dynamics from a single realistic video. We include them in our future work.

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

# Appendix

## A    Impact Statements

This work introduces a novel deep learning architecture tightly integrated with a classic mechanical analysis framework, the Discrete Element Method, to efficiently learn 3D particle dynamics from only 2D observations. The success provides insights into the combination of physics-augmented deep learning and 3D neural rendering for areas, e.g. physical simulations, engineering mechanical analysis, and computer graphics.

## B    Nomenclature

we list all symbols and marks used in the main paper in the following Table for the convenience of reference.

Table 5: Nomenclature. This table is split across pages

| Variable | Description |
|---|---|
| $i$ | $i$-th particle |
| $\mathbf{x}_i$ | position vector of particle |
| $\mathbf{v}_i$ | velocity vector of particle |
| $\mathbf{a}_i$ | acceleration vector of particle |
| $\mathbf{u}_i$ | movement vector of particle |
| $\mathbf{F}_{ij}^p$ | potential interaction force vector |
| $\mathbf{F}_{ij}^v$ | viscous force vector |
| $\mathbf{F}_i^g$ | gravity vector |
| $\mathbf{F}_{ij}^{pn}$ | potential interaction force vector along the normal direction |
| $\mathbf{F}_{ij}^{pt}$ | potential interaction force vector along the tangential direction |
| $\mathbf{F}_{ij}^{vn}$ | viscous interaction force vector along the normal direction |
| $\mathbf{F}_{ij}^{vt}$ | viscous interaction force vector along the tangential direction |
| $\delta d_n$ | intrusion scalar |
| $\delta d_t$ | instantaneous tangential displacement |
| $\Delta t$ | time interval |
| $r_i$ | radius of particle |
| $A_i$ | attribute of $i$-th particle |
| $F_{ij}^b$ | bond particle between particles which belong to the same object |
| $\mathcal{F}_c^n$ | mapping from intrusion to normal contact force |
| $\mathcal{F}_c^t$ | mapping from intrusion to tangential contact force |
| $\mathcal{F}_v^n$ | mapping from intrusion to normal viscous force |
| $\mathcal{F}_v^t$ | mapping from intrusion to tangential viscous force |
| $\mathcal{F}_b^n$ | mapping from intrusion to normal bond force |

| Variable | Description |
|---|---|
| $\mathcal{O}_k$ | set of particles belonging to the same $k$-th object |
| $\mathbf{v}_{ij}^t$ | the tangential velocity vector |
| $\mathbf{n}$ | normal unit vector |
| $\mathbf{t}$ | tangential unit vector |
| $\phi^n$ | the normal dissipative model |
| $\phi^t$ | the tangential dissipative model |
| $\Phi^n$ | normal graph kernel |
| $\Phi^t$ | tangential graph kernel |
| $\mathcal{H}_c$ | prediction head to regress the magnitude of the contact forces |
| $\mathcal{H}_b$ | prediction head to regress the magnitude of the bond forces |
| $\sigma$ | sigmoid activation function |
| $f_{ij}^{bn}$ | the normal bond force |
| $f_{ij}^{bt}$ | the tangential bond force |
| $f_{ij}^{cn}$ | the normal contact force |
| $f_{ij}^{ct}$ | the tangential contact force |
| $f_{ij}^{n'}$ | precomputed normal force vector |
| $\mathbf{f}_{ij}^n$ | normal force vector |
| $f_{ij}^{t'}$ | precomputed tangential force |
| $\mathbf{f}_{ij}^t$ | tangential force vector |
| $\mu$ | the frictional coefficient |
| $n_i$ | $i$-th node features |
| $v_i(t)$ | velocity of $i$-th particle at timestep $t$ |
| $h_i$ | latent embedding from i-th particle attribute |
| $e_{ij}$ | edge features between node $i$ and $j$ |

## C  Detailed architecture

We describe the detailed architecture used in the DEL. The entire graph operator is depicted in Equation 9 to Equation 16 in the main paper.

The embedding layer is an MLP with 2 hidden layers. The output embedding latent vector is normalized to [-1,1]. The $A_i$ and $A_j$ refer to the embedding vectors of the per-particle attribute, which we define as $A = [mat, d_o, ||a_i^{t-1}||_2]$. Here the $mat$ denotes the type of materials, $d_o$ is the distance between the particle to the mass center at the current timestamp, and $||a_i^{t-1}||_2$ represents the value of the acceleration at the last timestamp of this particle. In our learning framework, the $A$ mainly represents the specific material and its properties. During training, the properties of this material are encoded into the embedding of $A$.

The $\Phi^n$ and $\Phi^t$ aim to map related physical quantities to the abstracted node and edge features which we implement by graph neural networks. $\delta d_n$ in $\Phi^n$ is the initial edge features, we also consider it as

$e_{ij}$. Therefore, $\Phi_n$ can be described as:

$$
\begin{aligned}
temp_{ij}^l &= \psi_1(e_{ij}^{l-1}, h_i, h_j) \\
e_{ij}^l &= \psi_2(temp_{ij}^l, e_{ij}^{l-1}) \\
temp_i^l &= \frac{1}{\mathcal{N}} \sum_{\mathcal{N}(i)} e_{ij}^l \\
n_i &= Norm(\psi_3(temp_i^l, h_i))
\end{aligned}
\tag{19}
$$

where $temp$s are temporary intermediate variables. $n_i$ and $e_{ij}^l$ are the final output of the $\Phi_n$. $\psi_{1,2,3}$ are three 2-layer MLPs with residual connection. $e_{ij}^{l-1}$ is the edge feature from the last network layer. $Norm$ refers to learnable Layer Normalization. The reason why we perform an aggregation operation before computing the normal forces is the following. We assume that the mechanical behavior of a certain particle should be related to the external intrusion and the properties of its surrounding vicinity. Therefore, we use this graph aggregation to encode the information of its vicinity ($n_i$) and the external influence ($e_{ij}$). The abstracted features then are input to two different output heads $\mathcal{H}_b, \mathcal{H}_c$ to produce the final magnitude of the forces. The two heads are implemented as small MLPs with 2 layers and 200 hidden dimensions.

As for $\Phi_t$ in Equation 15, $\|v_{ij}^t\|_2$, $f_{ij}^{n'}$, and $e_{ij}$ are input edge features. In addition, $n_i$ and $n_j$ are the concatenation of the $n_i$ from Equation 10 and the $h_i$ from Equation 9 because we aim to emphasize the original particle attributes which affect the tangential force.

$$
\begin{aligned}
temp_{ij}^l &= \psi_1(e_{ij}, n_i, n_j) \\
e_{ij}^l &= \psi_2(temp_{ij}^l, e_{ij}^{l-1}) \\
temp_i^l &= \frac{1}{\mathcal{N}} \sum_{\mathcal{N}(i)} e_{ij}^l \\
f_{ij}^{t'} &= \psi_3(e_{ij}^l, temp_i^l, temp_j^l)
\end{aligned}
\tag{20}
$$

where $e_{ij} = cat([e_{ij}^{l-1}, \|v_{ij}^t\|_2, f_{ij}^{n'}])$, $n_i = cat([n_i^{l-1}, h_i])$. The final outputs are $f_{ij}^{t'}$ and $e_{ij}^l$. The rest of the architecture remains the same with Equation 20. Besides, we use a simple MLP with two hidden layers (each layer includes 200 neurons) to model $\phi^t$ and $\phi^n$. Also, the sigmoid activation is used before outputting the coefficients because the viscous forces are manifested as a reduction in potential interaction forces. After $\mathbf{f}_{ij}^t$ in Equation 16 and $\mathbf{f}_{ij}^n$ in Equation 14 are obtained, we aggregate the forces for each particle:

$$
\mathbf{f}_i = \sum_{\mathcal{N}(i)} (\mathbf{f}_{ij}^n + \mathbf{f}_{ij}^t)
\tag{21}
$$

Thus we can update their velocities and positions by the Euler integration:

$$
\begin{aligned}
\mathbf{v}_t &= \mathbf{v}_{t-1} + \frac{\mathbf{f}_i}{m_i}\Delta t \\
\mathbf{x}_t &= \mathbf{x}_{t-1} + (\mathbf{v}_{t-1} + \frac{\mathbf{f}_i}{m_i}\Delta t)\Delta t
\end{aligned}
\tag{22}
$$

## D   Implementation Details

In this section, we introduce the implementation details of our experiment setup. We first use a pretrained GPF to initialize the scene as particles.

For the dynamic module, we build the graph at each timestamp via the k-nearest neighbor search with a fixed radius of 0.025. The particle radius ($r$ for different materials) is set to equal the search radius initially and will be optimized through the training process to be a property of the material. In addition, we place a heavy rigid table at the bottom of each scenario, which is also represented by particles but we do not update its position, and its velocity is constantly set to zero. The only function of it is to support the moving objects above. Additionally, the mass parameter for each particle is set to initialize at 1 and optimized during the training phase as well. All models are trained via AdamW optimizer with 5e-4 learning rate. We adopt the StepLR schedule to adjust the learning rate with the increasing step. After 10,000 iterations, we multiply the learning rate by 0.9.

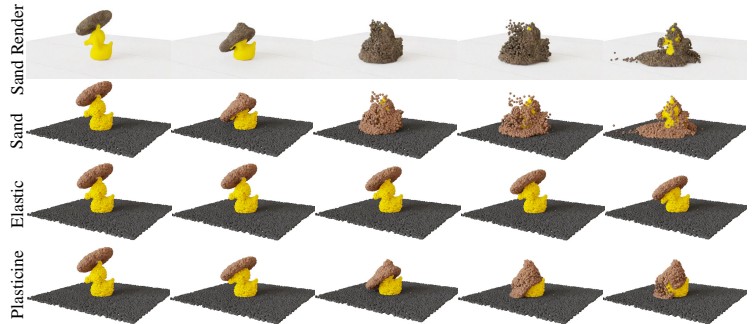

Figure 7: Examples about material swapping on SandFall

# E    Swapping Materials.

Our method has a unique advantage. We can swap materials used in simulation by swapping the material embeddings and graph kernels because different materials share the same mechanical framework and the graph network in it is only responsible for mapping physical quantities and material embeddings to the forces. We show an example to illustrate this application. As shown in Figure 7, the first and second rows are the predicted rendered views and particle views of the SandFall respectively. We change the graph kernel and material embedding of the sand to the elastic and plasticine ones respectively. We can observe corresponding changes in the mechanical behavior of particles.

# F    Detailed Dataset Description

In this section, we thoroughly describe the data generation process for our experiments, employing the Material Point Method (MPM) to simulate interactions among various materials. We designed six distinct scenarios to encompass a diverse range of material combinations: Plasticine, Multi-Objs, Bear, FluidR, SandFall, and Fluids. Each scenario is represented through 128 dynamic episodes, differentiated by unique initial conditions such as shapes and velocities, while maintaining consistent material properties, including elasticity and viscosity coefficients, within each scenario.

For the reconstruction of meshes from simulation data, we utilized SplashSurf, followed by rendering multiview dynamic image sequences using Blender. At every timestep, the motion is captured from four distinct camera angles. The objective of this research is to derive material interaction behaviors from these 2D observational sequences.

**Plasticine.** This scenario features interactions between two distinct materials: a red elastic object and a blue elastoplastic material. We initialize the elastoplastic materials in various shapes, including duck and bunny configurations, which are then subjected to impacts from the red elastic ball from multiple directions.

**Multi-Objs.** This scenario encompasses interactions among five objects composed of three distinct materials. The blue ball and cuboid are categorized as rigid. In contrast, the cylinder and rainbow-colored ring are elastic, each characterized by unique values of elastic modulus and Poisson's ratio. The experimental setup initiates with the rigid ball colliding with the rigid cuboid, triggering a sequence of subsequent collisions. Each episode is distinguished by varying arrangements and initial configurations of the objects, showcasing diversity in shape and positioning.

**Bear.** This scenario investigates interactions among objects composed of three distinct materials: plastic, elastic, and rigid. The setup includes a brown bear modeled as a plastic toy, a triangular ship constructed from an elastic material, and a heavy, rigid, yellow box. The experimental design involves the ship and box colliding with the plastic bear from various directions and positions, aiming to study the resultant material behavior and object interactions under different impact scenarios. Each collision is designed to explore the dynamic responses of plastic, elastic, and rigid materials when subjected to varying forces and angles of impact.

**Fluids.** The Fluids scenario examines the dynamics of fluid behavior as it impacts a tabletop, with the fluid initially shaped into complex geometries, including forms reminiscent of a bunny, Pokémon, or duck, each varying in size. This setup facilitates observations of the fluid's response upon collision

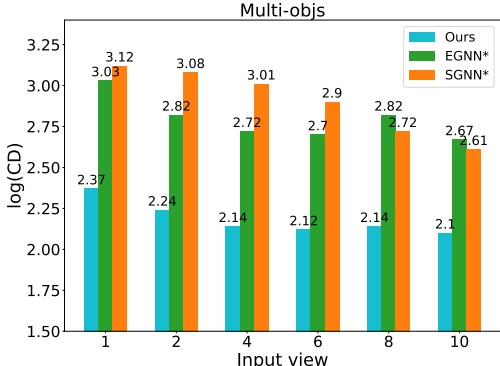

Figure 8: Experiments of model evaluation on the SandFall with different numbers of input views. The x coordinate is the number of input views, and the y coordinate is the log Chamber Distance.

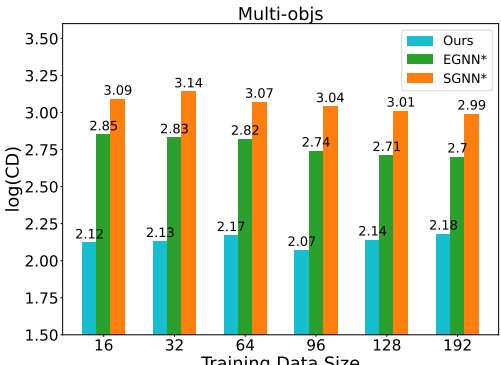

Figure 9: Experiments of the number of training episodes on Multi-objs dataset. The x coordinate is the number of training data, and the y coordinate is the log Chamber Distance.

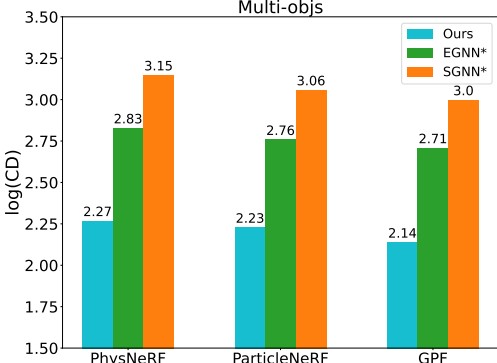

Figure 10: Particle-view results of the dynamic prediction on Multi-objs dataset by using different particle-based renderers. The y coordinate refers to the log Chamber Distance metrics.

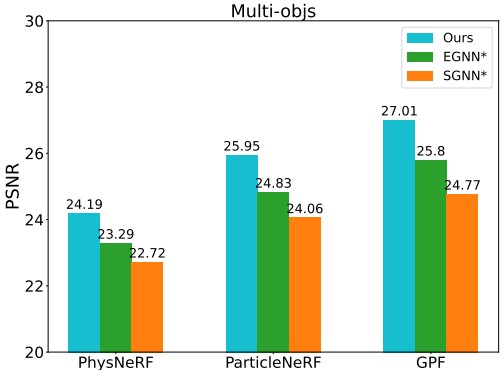

Figure 11: Rendering results of the three methods on Multi-objs dataset by using different renderers. The y coordinate refers to the PSNR metrics.

with the ground, encapsulated within a scenario that includes invisible boundaries to constrain the fluid's spread. Upon contacting these boundaries, fluid particles alter their trajectories, enabling detailed study of fluid dynamics and boundary interactions. This scenario allows for the exploration of fluid behavior under varied initial conditions, contributing to our understanding of fluid dynamics in controlled environments.

**FluidR.** In this scenario, we explore the dynamics of liquid flow over a hollow shelf with a complex geometry. Across different episodes, we systematically vary the initial configurations of the liquid, altering both its shape and the height from which it is released atop the shelf. The shelf itself is constructed from a hard elastic material, characterized by a very high Young's modulus, to study the interaction between the liquid's fluid dynamics and the shelf's structural response. This setup provides a unique opportunity to observe the behavior of liquids in contact with elastic materials under varying initial conditions, offering insights into fluid-structure interaction phenomena.

**SandFall.** This scenario investigates the dynamic interaction between a life buoy, composed of sand soil, and an elastic toy duck. Upon impact, the toy duck undergoes deformation and subsequently recovers to its original shape, causing the sand soil life buoy to rebound. A significant challenge in modeling the dynamics of this interaction lies in the partial obscuration of the toy duck by the sand soil during impact. This obscuration results in incomplete information regarding the duck's deformation and response, complicating the task of accurately learning the system's dynamics. The study focuses

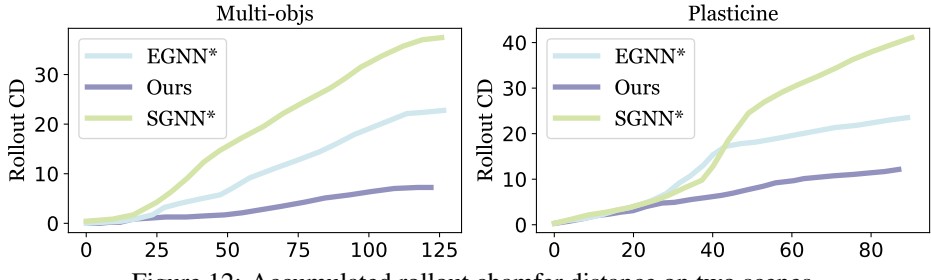

Figure 12: Accumulated rollout chamfer distance on two scenes

on understanding and overcoming the limitations imposed by insufficient visual information on the elastic response of the toy under the impact of granular material.

# G    Additional Results and Comparisons

## G.1    The establishment of Rollout metrics

We in addition show two line diagrams of the accumulated Rollout Chamfer Distance in the two sequences in Plasticine and Multi-Obj, to further indicate the superiority of our dynamics module (Fig. 12).

## G.2    The influence of different renderer

The default renderer utilized in the main paper is the GPF. In this section, we evaluate our methods and the baselines with the other two particle-based renderers, i.e. PhysNeRF [13], ParticleNeRF [41], both of which can render dynamic particles. PhysNeRF is employed in Neurofluid to represent and render scenes. However, in the original paper, it is jointly trained with the dynamics module and requires more camera views. For a fair comparison, we pretrain the PhysNeRF from scratch and freeze its parameters for all models. In addition, we provide an initialized particle set for PhysNeRF and ParticleNeRF because they are difficult to initialize the scenes as point clouds from sparse camera views (4 in our experiments), while GPF is capable of initialization due to its depth estimation module. Figure 11 and Figure 10 show the rendering and dynamic prediction results of different renders. We can see our method achieves better performance regardless of which renderer is employed. Moreover, rendering capability scales proportionally with the accuracy of learned dynamics. As GPF produces better rendering quality, more accurate dynamics can be learned by utilizing it. This example illustrates that our method is robust to the renderer selection because physical priors always guide the model toward a reasonable learning target.

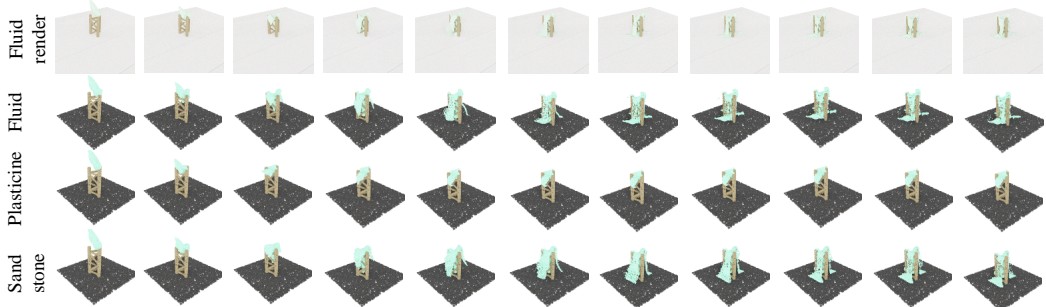

Figure 13: Detailed example of swapping materials without retraining the model on the FluidR scenario

## G.3    The influence of the number of camera views in training

In this section, we use various numbers of camera views to train the models. In the main paper, four camera views are deployed across four corners of the scene. Here we additionally evaluate if 1, 2, 6, 8, 10 cameras are used, what will happen? The ablation results on Multi-Objs dataset are reported

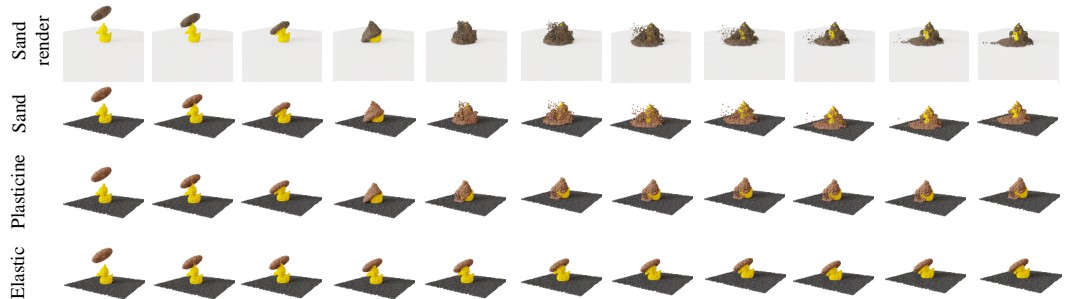

Figure 14: Detailed example of swapping materials without retraining the model on the SandFall scenario

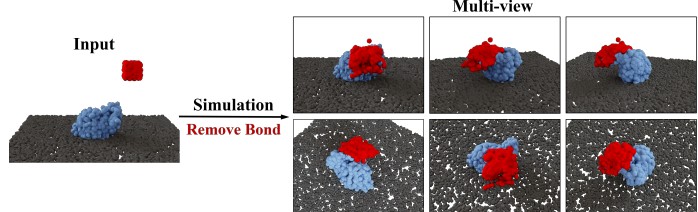

Figure 15: Visualization of the case to remove the bond force.

in Figure 8. It is observed that our approach is not sensitive to the number of cameras, even if only one camera is used, due to the physical constraints posed by our mechanics framework. The search space is contracted by the physical knowledge injected into the model. Therefore our method can be effectively trained via only visible particles while the invisible ones are constrained through physics. On the contrary, the other two baselines are severely affected by the number of cameras. They cannot learn well from sparse cameras. With the increase in the number of cameras, their performance also improved and the upward trend shows no signs of abating. We can assume that the maximum upper bound of such improvement should be the performance when the 3D labels are used to train them.

### G.4 The influence of different sizes of training samples

We evaluate how the training set size affects the performance and the results are reported in Figure 9. With increasing sizes of the training set, the log Chamber Distance of EGNN* and SGNN* experience slight rises. However our performance remains stable and always stays on top, which means our method is not sensitive to the training sizes and can learn from sparse data.

### G.5 What will happen if we remove the bond force?

In this subsection, we claim the importance of the bond force in the simulation system and the significance of using two independent networks to model contact and bond force separately. We did two additional experiments. The first is that we remove the bond force from the entire system, i.e. remove Eq. 12. The second is that we use a single network to predict the bond force and contact force simultaneously, i.e. merge Eq. 12 and 11 together. We observe that the simulator cannot keep the original shape of rigid and elastic bodies in either case. We show a visual example in Fig. 15 and Fig. 16 to illustrate this. These two experiments prove the effectiveness and necessity of this design to separately model the two forces by distinct networks.

### G.6 How does the performance when 3D groundtruth particle labels is used

We here report the evaluation of training these models with 3D particle tracks. Even though our approach aims to learn 3D dynamics from images, the mechanics-encoded paradigm is also helpful for learning from 3D labels. Table 6. show the rollout mean square errors of the three models in all scenarios. From this comparison, SGNN's performance has rapidly risen and is roughly on par

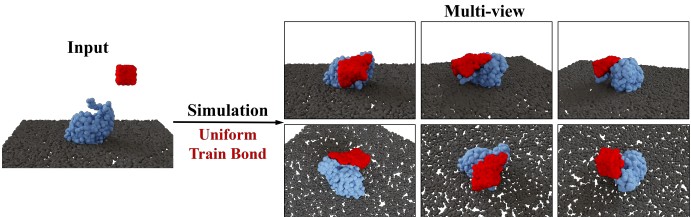

Figure 16: Visualization of the case to jointly model the contact and bond force by a single network.

Table 6: Quantitative comparisons between our method and benchmarks on five scenarios in particle views.

| | Given 3D GT Rollout MSE $\times$ 10^2 | | | | | |
| Method | Plasticine | SandFall | Multi-Objs | FluidR | Bear | Fluids |
| --- | --- | --- | --- | --- | --- | --- |
| Ours | **0.275** | **0.0414** | **0.239** | 0.136 | 0.301 | **0.051** |
| SGNN* | 0.268 | 0.0468 | 0.252 | **0.121** | **0.282** | 0.062 |
| EGNN* | 0.614 | 0.127 | 0.568 | 0.318 | 0.705 | 0.143 |

with our method. Both of them outperform EGNN by a considerable margin, which is consistent with that reported in their original paper. This further proves that the reason why SGNN fails to perform well under pixel supervision is caused by the uncertainty of 2D to 3D. EGNN is better than SGNN under 2D supervision because it predefines the direction of message passing which reduces the learning space as well. More importantly, our promising performance illustrates the effectiveness of incorporating strong mechanical priors for both 2D and 3D labels used. Our approach seems to excel in simulating solids, particularly elastic and rigid bodies. However, the SGNN method outperforms in simulating particulate matter like sand and viscous liquids. This discrepancy may stem from the presence of bond forces in our mechanical framework, constraining particles belonging to the same material.

### G.7    Additional Demonstrations of Swapping Materials

Our methodology introduces a novel capability for dynamic material pair substitution within pre-existing simulation environments. By predefining mechanical responses and employing a GNN solely for mapping deformations to interaction forces, we facilitate the modification of material interactions with minimal adjustments. This process involves substituting the parameters of the GNN kernel with those derived from alternative scenarios and altering the input $A_{ij}$, representing the adjacency matrix or interaction terms.

Figure 13 illustrates this concept by substituting the original fluid-elastic interaction pair with a plastic-elastic pair in the Bear scenario, and a sand-elastic pair in the SandFall scenario, demonstrating the adaptability of our approach to simulate varied mechanical behaviors. Similarly, Figure 14 presents another application of our method, where the sand soil material is replaced with elastic and plastic materials, leveraging GNN kernels trained in distinct contexts. These examples underscore our method's versatility in simulating diverse material behaviors through strategic parameter adjustments.

## H    Visualization of Learned Particle Interaction Forces

In this section, we visualize the contact forces and bond forces to validate the physical meaning and interoperability of the learnable GNN kernel. After training the DEL, the $\phi^n$, $\mathcal{H}_c$ and $\mathcal{H}_b$ in Eq. 10, 11, 12 should correctly map intrusion into the contact and bond force magnitudes. We extracted these GNN kernels from the simulation system separately to evaluate their responses and outputs to different intrusion inputs. For simplicity, we only evaluate the normal direction. The recorded results are visualized in Fig. 17. In this figure, the x-axis refers to the intrusion value ($\delta d$ in the paper), the y-axis denotes the normalized output of Eq. 11, and 12. The solid line denotes the contact force and the dashed line denotes the bond forces. We represent different materials in various colors. We can see from this figure that for rigid bodies, even very small displacements can result in significant resistance, preventing the object from deforming. For sand and water, since they do not need to

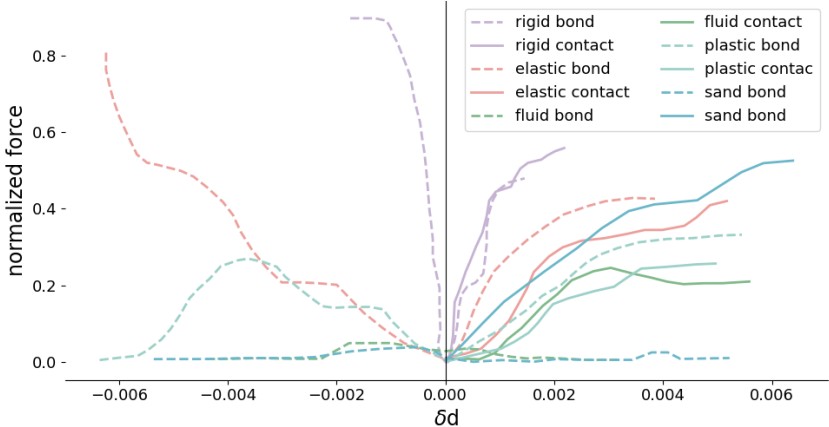

Figure 17: Visualization of the learned constitutive mapping. The x-axis refers to the intrusion and the y-axis denotes normalized force magnitude.

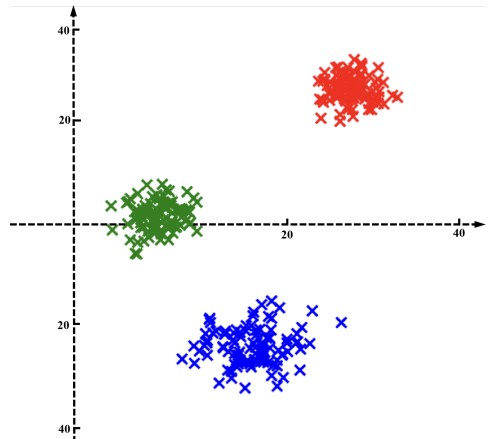

Figure 18: Visualization of the learned material embeddings on Multi-Obj scenes.

maintain original shapes, our network adaptively learns to set the bond forces to very small values. For plastic materials, our network has also learned to break the bond links at appropriate times (the bond forces are close to 0). Through this visualization, we would like to claim that our GNN kernel has indeed learned real physical meaning i.e. the forces between particles for simulating different materials.

# I  Visualization of Learned Material Embeddings

Furthermore, following the suggestion of the reviewer, we visualize the learned feature vector, i.e. $h_i$ in Eq. 9, by using the t-SNE method, which is shown in Fig. 18. In this figure, the red scatter points are projected by the feature of Rigid particles. The green and blue scatter points are produced by the feature of particles belonging to two different types of elastic bodies. The features belonging to the same object have clustered together. The reason they do not overlap completely is because their positions relative to the object are also encoded in the $A_i$. As shown in the Figure, the red points are closer together, which is because for rigid bodies, regardless of where the points are on the object, they have a strong resistance to deformation. We believe this visualization demonstrates that our framework has learned the material-specific features.

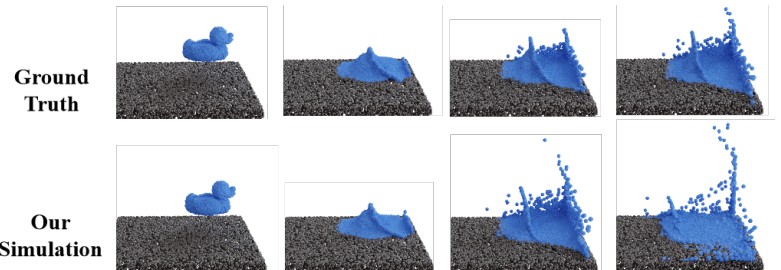

Figure 19: Failure examples of the proposed method when simulating non-Newtonian fluids.

## J   More Qualitative Results in both render and particle views

In this section, we report additional qualitative results from Fig. 21 to Fig. 30. These figures include both particle view and render view of our method, EGNN*, VPD, and 3DIntphys (only available for particle views). More detailed results can be seen in our supplementary video. For the EGNN* and our method, we demonstrate both render and particle views to faithfully compare the predicted dynamics.

It can be observed from these figures that our method can generalize well to different initial shapes and conditions, especially when predicting long-term dynamics. While the VPD and EGNN* cannot precisely predict the mechanical behaviors between materials. Or they only produce plausible results on some certain materials. For instance, in some cases, such as BEAR and FluidR, the EGNN* can deliver plausible results at the early stage of interactions, but the performance degrades drastically with the progressing deformation.

We in addition show a comparison of baselines for the Fluids dataset in Figure 20. Our method preserves the basic trend of water flow, which other methods do not.

## K   Failure Cases

In this section, we discuss failure cases of the proposed method. We observe that this method struggles to simulate non-Newtonian fluids because a fundamental assumption of this method is that the material properties are consistent throughout the training dataset. However, the properties of non-Newtonian fluids change with stress variation. Here we show an example in Fig. 19. If we use our method to simulate non-Newtonian fluid, a large gap between the prediction and groundtruth can be observed.

This method is also not suited to simulate smoking. First, smoke consists of extremely small water vapor particles, with the particle size being very small and the number of particles being vast. Simulating smoke by particles requires significant computational resources. Second, smoke simulation not only involves the exchange of momentum between particles but also is governed by thermodynamics, making the simulation of smoke with particles a complex topic.

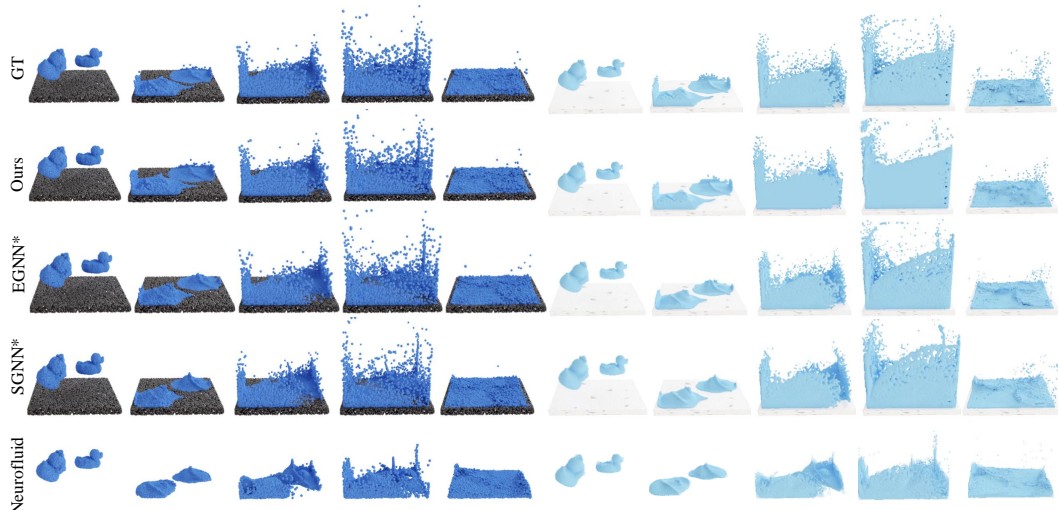

Figure 20: Qualitative Comparisons of all baselines on Fluids dataset in both rendering and particle views.

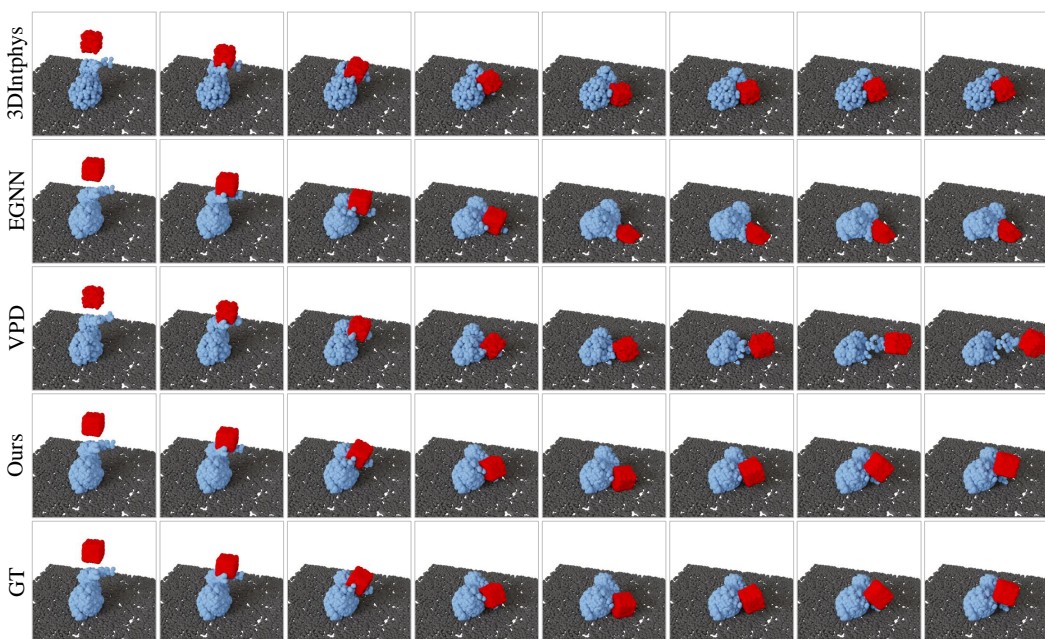

Figure 21: Long-term predictions of 3DIntphys, VPD, EGNN and our method on Plasticine scenarios in particle view.

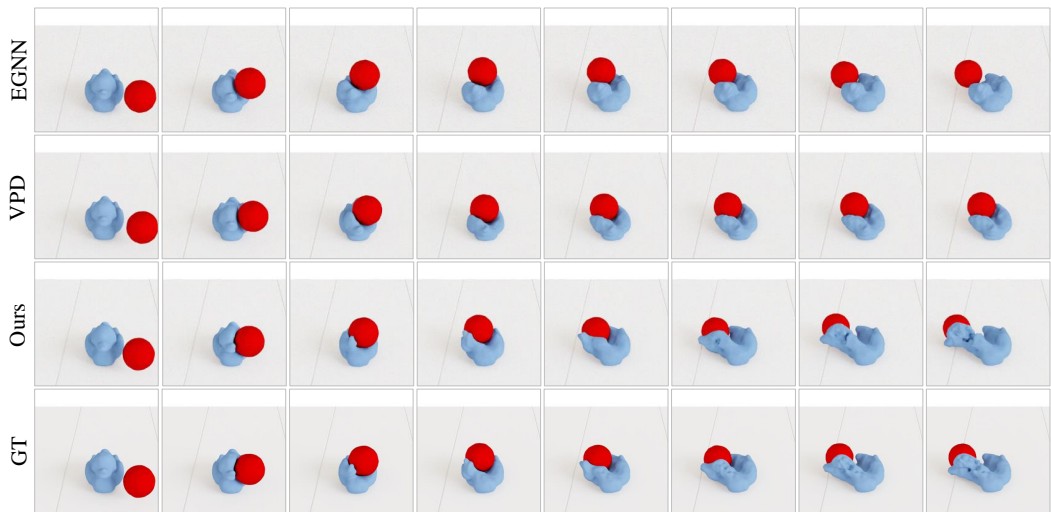

Figure 22: Long-term predictions of VPD, EGNN and our method on Plasticine scenarios in render view.

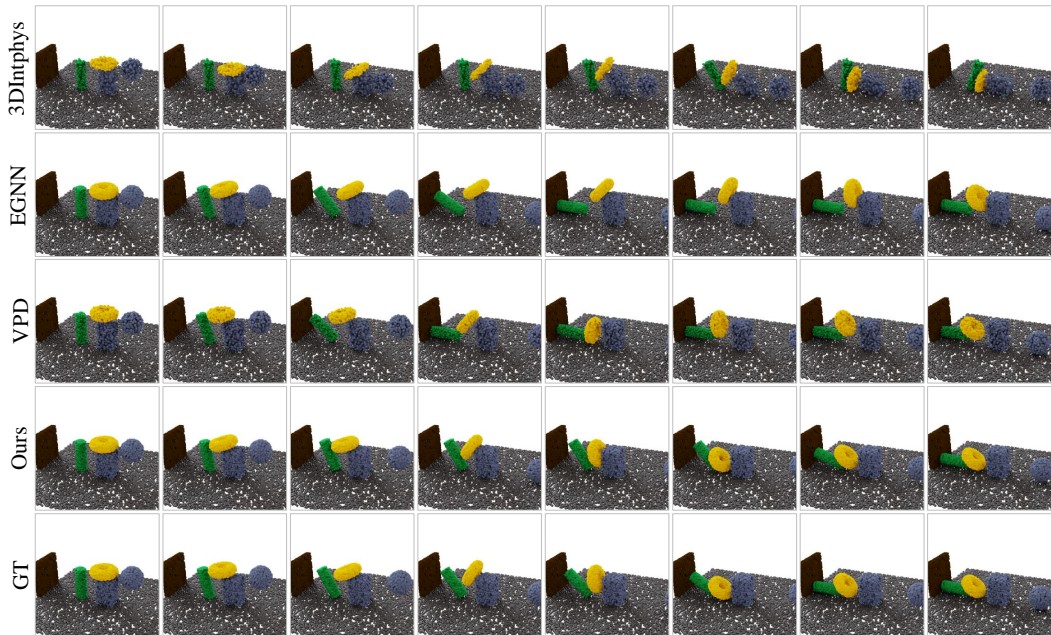

Figure 23: Long-term predictions of 3DIntphys, VPD, EGNN and our method on Multi-objs scenarios in particle view.

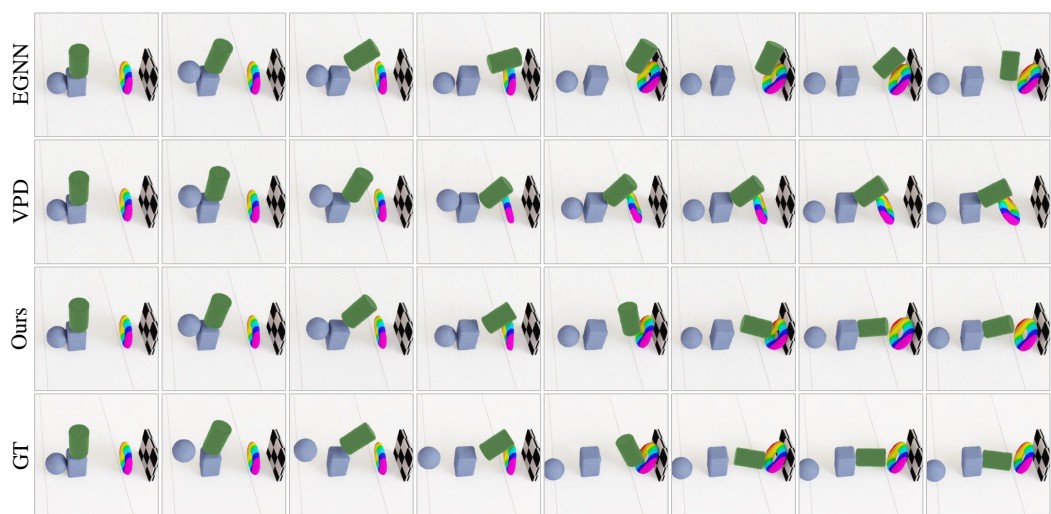

Figure 24: Long-term predictions of VPD, EGNN and our method on Multi-objs scenarios in render view.

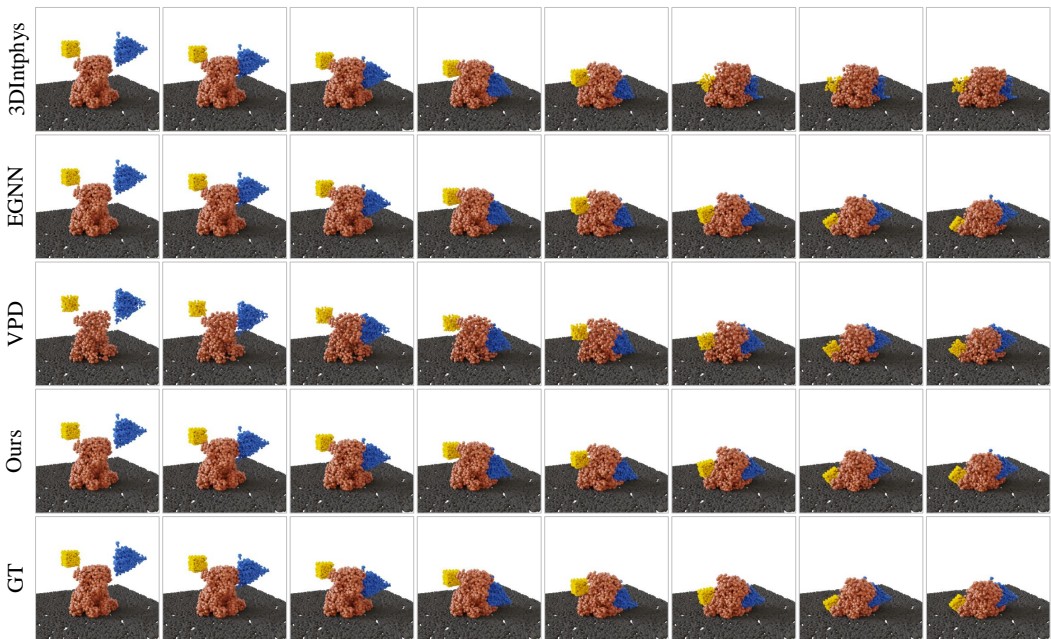

Figure 25: Long-term predictions of 3DIntphys, VPD, EGNN and our method on Bear scenarios in particle view.

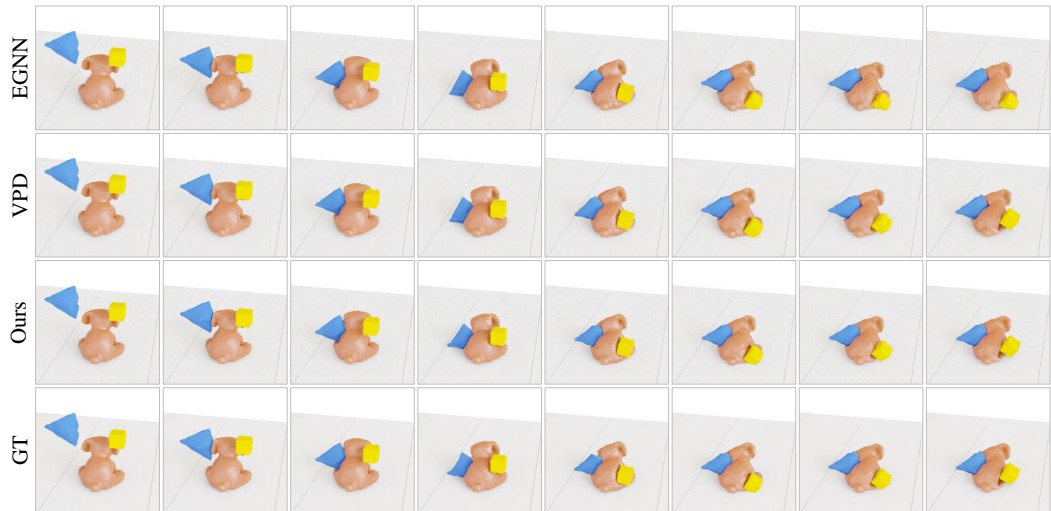

Figure 26: Long-term predictions of VPD, EGNN and our method on Bear scenarios in render view.

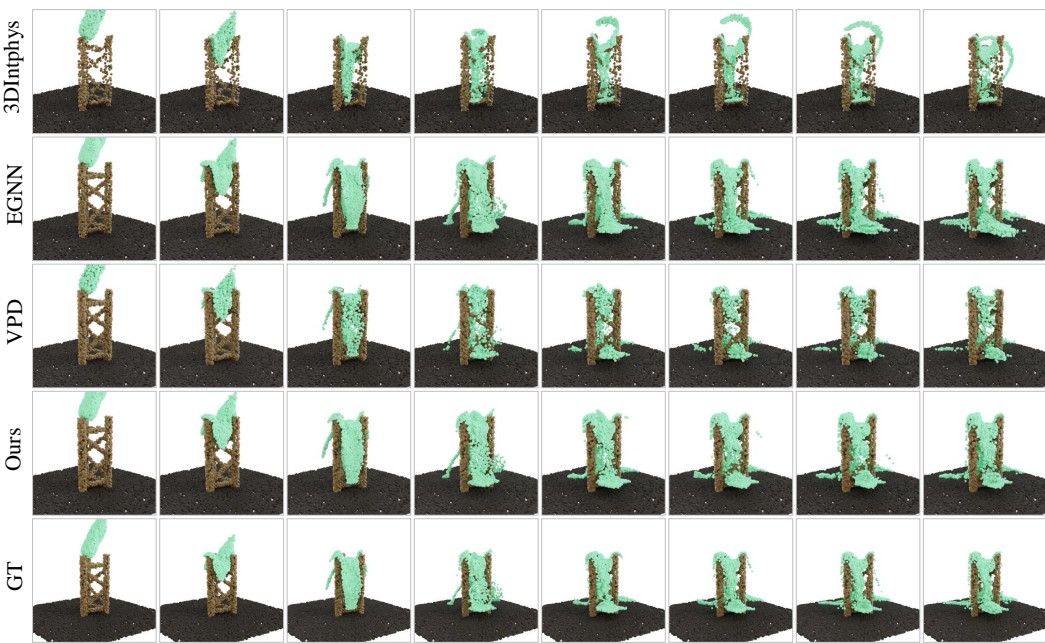

Figure 27: Long-term predictions of 3DIntphys, VPD, EGNN and our method on FluidR scenarios in particle view.

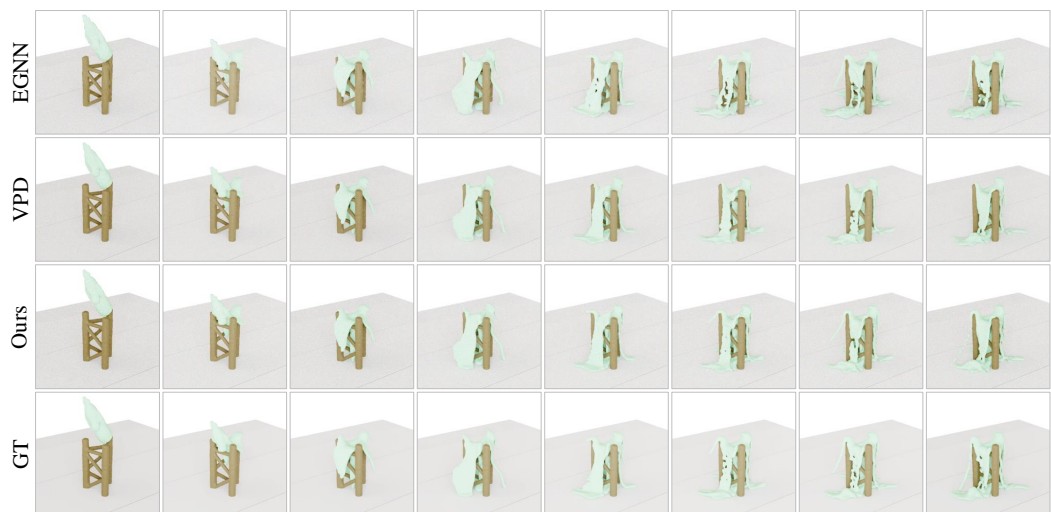

Figure 28: Long-term predictions of VPD, EGNN and our method on FluidR scenarios in render view.

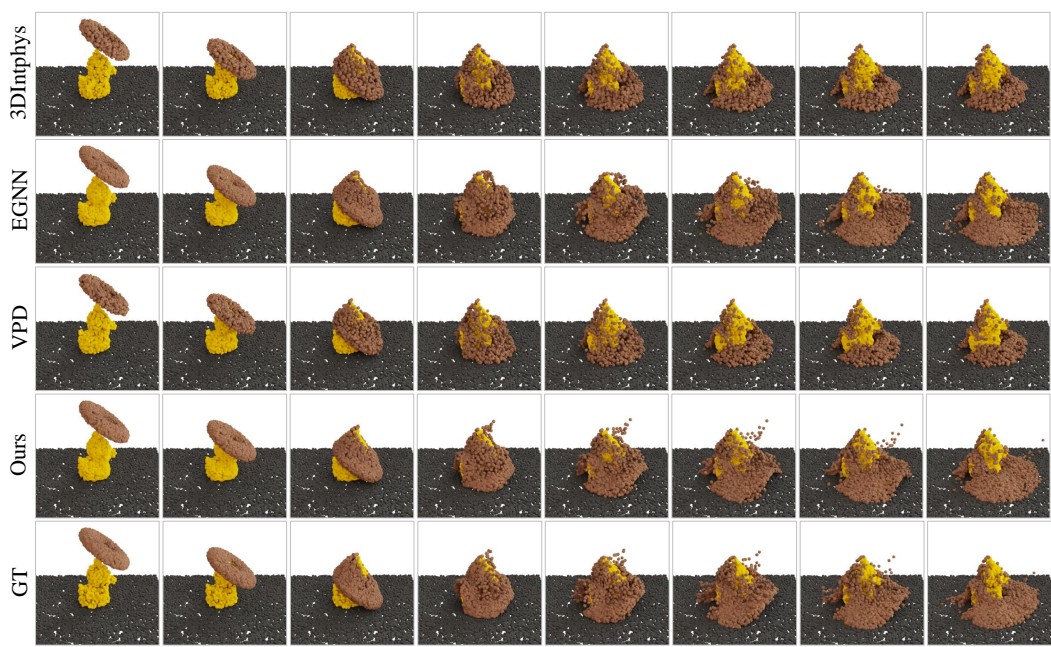

Figure 29: Long-term predictions of 3DIntphys, VPD, EGNN and our method on SandFall scenarios in particle view.

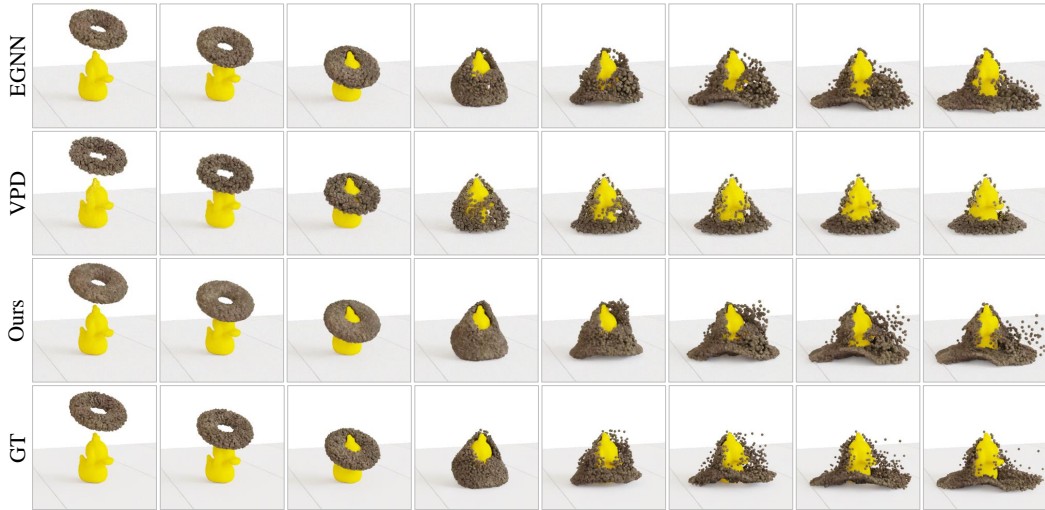

Figure 30: Long-term predictions of VPD, EGNN and our method on SandFall scenarios in render view.

