# OpenReview forum: "DEL: Discrete Element Learner for Learning 3D Particle Dynamics with Neural Rendering"
_NeurIPS.cc/2024/Conference — NeurIPS 2024 poster_

### Official Review · Reviewer_uaqu · 2024-06-27

**Soundness:** 3
**Presentation:** 3
**Contribution:** 3
**Rating:** 7
**Confidence:** 3

**Summary:**

This paper presents a new method to learn 3D particle dynamics from sparse 2D observations using inverse rendering.
In constrast to previous work, it does not learn a fully unconstrained model but makes use of known physical priors.
The method learns graph network kernels to model the particle interation forces in the DEA framework, which are classically designed domain experts.
These graph networks are trained from 2D observations using a differentiable renderer without any 3D supervision.
The method is evaluated on different scenes and using different physical materials.
It surpasses the provided baseline comparisons in the test shown in the paper.

**Strengths:**

The paper presents a novel combination of graph networks and classical particle-based simulation that, in conjunction with a differentiable renderer, allows it to recover 3D particle dynamics from images.

The presented method is technically sound, well explained, and easy to follow.
The evaluation is sensible and very extensive, encompassing different material configurations, and shows a siginficant improvement over the SoTA.

The source code and dataset will be made available to the public.

**Weaknesses:**

There are several things that are not clear to me, especially related to scene initialization and the initial velocities. These points are detailed in the questions below. I don't think these are fundamental problems, but I nonetheless hope the authors can clarify these aspects in the rebuttal.

**Questions:**

- 3.3: the $v_{ij}^{n}$ or $v_{ij}^{t}$ used in equations 14 to 16 do not match any of the definitions in appendix B.
- For training (3.4), how is the scene initialization done with the renderer? And how accurate is the estimated initial velocity based on this initialization?
- Closely related: how sensitive is your method to inaccuracies in the initial velocity?
- How long are the sequences used in your experiments?
- How long is your training (number of iterations, epochs, or variable updates)?
- Are the views shown in the figures the ones that were used for training or are they novel views?

There are several grammatical errors throughout the paper, for example:
- 291 "metrix"
- 291 "cham**b**er distance" should be "cham**f**er distance"
- 306 There seems to be a new paragraph missing before "Results in particle view"
- 311 "we claim the reason that..."
- 343 "while keep the priors regulated..."
I would encourage the authors to do a complete pass over the paper to fix any writing mistakes.
The tables and figure on page 9 are also very squeezed.

**Limitations:**

Limitations are briefly discussed.

---

> ### Author Rebuttal · Authors · 2024-08-03
>
> > **Weakness and Question 2 and 3: Scene and Velocity Initialization**
>
> Thanks for the reviewer pointing out these ambiguities. We explain these details in the following and will add them to our revised manuscript for better readability.
>
> **Scene Initialization.** As we stated in the first paragraph of *Section 3*, we adopt the Generalizable Point Field (**GPF**) [1] to initialize the scene. GPF is a point-based NeRF-like approach, which can directly convert multiview images into a point-based NeRF representation in a generalizable way because GPF is pretrained on large 3D reconstruction datasets. In detail, GPF first projects 2D images into 3D points by predicting their depth maps. Second, it hierarchically aggregates features from images to the point scaffold to obtain separate appearance and geometry features. GPF can render changeable content by moving these featured points. We slightly fine-tuned the GPF on our training set based on the authors' provided checkpoints for better reconstruction.
>
> Moreover, we also evaluate some other point-based renderers in *Section G2, Figures 10 and 11* in the Appendix, the results illustrate that **DEL is also robust to these renderers**. We would like to claim that all point-based renderers can be adopted to train DEL as long as **(1)** they can render different content with the point movement **(2)** they are fully differentiable to propagate gradients.
>
> **Velocities Initialization.** We follow the methods in PAC-NeRF [2] and PhysGaussian [3] to use images by using the first three frames to optimize the initial velocities. We assume that the initial velocities for all particles of the moving objects are identical. Hence, we only need to run the following loop:
> 1. optimize the velocity,
> 2. move the particle set according to it,
> 3. render novel views based on the new position of the particles
> 4. compute the loss between the rendered views and the real views
> 5. update the velocity to align the renderings with the real views
>
> After the error is smaller than a predefined threshold, the loop is broken.
> Following the reviewer's advice, we additionally evaluate the effect of the approximated velocities. We train two models on the Plasticine scene using estimated velocities and another using actual velocities. Then, we test each model on two test sets, one provides the real velocities and another does not. The results are shown below:
>
> | Test/Train | w | w/o |
> |-------|-------|-------|
> | w | 7.37 | 7.48 |
> | w/o | 7.82 | 7.54 |
>
> This table reports the Mean Rollout Chamfer Distance for each situation. "w" refers to "with real velocities" and "w/o" refers to "without velocities".
> From the Table, we can see that the results do not exhibit significant fluctuations, which may be due to the simplistic nature of the initial velocity estimations. How to estimate initial velocities in a more complex environment remains a worthy topic for future research.
>
> *[1] Wang J, et al.. Learning robust generalizable radiance field with visibility and feature augmented point representation. International Conference of Learning Representations (ICLR) 2024*
>
> *[2]Li X, Qiao Y L, Chen P Y, et al. Pac-nerf: Physics augmented continuum neural radiance fields for geometry-agnostic system identification.  International Conference of Learning Representations (ICLR) 2023*
>
> *[3]Xie T, Zong Z, Qiu Y, et al. Physgaussian: Physics-integrated 3d gaussians for generative dynamics.  Conference on Computer Vision and Pattern Recognition. (CVPR) 2024*
>
> > **Question 1: Notation**
>
> We thank the reviewer for the careful check very much. The $v_{ij}^n$ in Equation 14 represents the relative velocity of particle i with respect to j, actually, the velocity could be a vector, so it should be modified as  **v**$_{ij}^n$ . (bond font represents a vector)
>
> Similarly, the $v_{ij}^n$ which denotes the tangential velocity should be modified as **v**$_{ij}^t$. We thoroughly check all notations in the paper and correct them all, and we update the Nomenclature in Appendix B accordingly.
>
> > **Question 4: The length of the sequences**
>
> For different simulation scenarios we have different sequence lengths. We report the simulation steps (i.e. the sequence lengths), which are averaged over the entire test set for each scenario, in the following Table.
>
> | Scenario | plasticine | SandFall | Multi-Obj | FluidR | Bear | Fluids |
> |-------|-------|-------|-------|-------|-------|-------|
> |Avg Step Number | 86 | 94 | 132 | 155 | 124 | 138 |
>
> To better clarify this, we add this description to Section F in our revised manuscript.
>
> > **Question 5: Training time**
>
> Thanks for this comment. As we stated in the answer of Weakness, we first finetuned the GPF by using their default learning rate on our training set for 500 iterations, this step only takes 12 minutes. Then we train our model on a certain scenario, for example the Plasticine,  for about 24000 iterations on a single NVIDIA RTX3090, and the loss has been steady.  The training time for a certain scenario is about 3.6~4 hours. We add the above training information to the *Section D* in our revised paper for the convenience of readers.
>
> > **Question 6: Test views**
>
> Thanks for this question. We would like to clarify that all camera views and initial conditions in the test set have never been seen in the training set. Because we believe that this should be better for evaluating the generalization ability of these models. We add this illustration in the first paragraph of Section 4 in our revised paper for better readability.
>
> > **Question 7: grammar errors and layout**
>
> We appreciate a lot the reviewer for pointing out these typos and errors. Following these suggestions, we correct them in the revised version. Moreover, we carefully review and improve the text's language and grammar, and optimize the layout to boost the quality of the paper. Now the revised version is more readable and clearer.

---

> > ### Comment · Reviewer_uaqu · 2024-08-12
> > **Re: Rebuttal by Authors**
> >
> > Thank you for the clarifications. Trusting that these unclear parts will be clarified in a final version I'd be happy to still support an "accept" for this paper.

---

> > > ### Author Response · Authors · 2024-08-14
> > >
> > > Thank you for these valuable comments and encouraging feedback. We are sure that they improve the quality of our manuscript by a large margin. Thanks again for your time and attention in the review and discussion periods.

---

### Official Review · Reviewer_fjMr · 2024-07-11

**Soundness:** 2
**Presentation:** 2
**Contribution:** 3
**Rating:** 5
**Confidence:** 4

**Summary:**

This paper proposes to incorporate the neural network with Discrete Element Analysis framework for particle-based simulation. The method adopts GPF as differentiable renderer and is trained through multi-view videos. The proposed model delivers faithful rollouts and outperforms the baselines.

**Strengths:**

* The proposed method deeply integrates the DEA and obtains robust performance.
* The model can be trained through multi-view videos in an end-to-end manner.

**Weaknesses:**

1. To simulate different scenes, such as deformable objects and fluid, it seems that one has to train different models on different cases, indicating scene-specific design with limited generalisation abilities.
2. The results of verifying the generalisation abilities are missing. For example, after trained on the deformable bunny as shown in Figure 5, can the model simulate objects of other shapes, such as a car, or objects composed of more particles?
3. The method seems to be capable of dealing with objects with limited particles and struggle to simulate large amount of particles.

**Questions:**

1. To enable the training on different scenes through the images, are different pertained GPF needed for different scenes?
2. How is the performance of the method for long-term predictions? For example, a trajectory with 150 frames.
3. At L253, the author claims to use L2 loss. However, for equation 17, it seems the loss is L1. Is this a typo?

**Limitations:**

Please refer to the weaknesses and limitations.

---

> ### Author Rebuttal · Authors · 2024-08-03
>
> > **Weakness 1: Scene Specific**
>
> Thanks for this comment. The proposed approach indeed needs to be trained for modeling different materials. In fact, we prefer to describe it as **material-specific** training rather than scene-specific training. This is because DEL can implicitly encode the mechanical properties of a certain material into the graph neural operator during training, and after training, the trained graph neural operator can be directly inserted **into any unseen scenes with different initialization conditions**, such as shapes and velocities.
>
> We also conducted additional experiments and analysis for this point in *Section E, Figure 7, Figure, 13, and Figure 14* in Appendix. It can be observed that various materials can be simulated by **simply exchanging the graph operators trained in different scenes**. We believe that this also shows the generalization ability of the DEL. An ideal situation to use this method is that one can train the graph kernel for a specific material in a certain dataset in advance, and then invoke it when one needs to simulate the material.
>
> > **Weakness 2: Generalization Ability**
>
> We would like to argue that all visualized results in our paper are from the **test set**, including the bunny in Figure 5 and the duck in Figure 22. Both of the initial shapes have never been seen by the model during training. As we stated in the *Dataset* part in Section 4, for each specific material pair, for example, the plastic and the rigid pairs in the **Plasticine** scenario (Figure 5), there are 128 different training sequences and 12 testing sequences. All of them have different initial shapes, velocities, and locations. We would claim that **the "Plasticine" scenario does not refer to the specific "bunny" or "duck", it refers to the same plastic material that makes up them**.
>
>  For example, in the training set, the plasticine might be initialized as a "sphere" or a "dog", but in the test set, it can be a "bunny" or a "duck". Therefore, we could claim that our model can be generalized well to different initial conditions that have never appeared in the training phase as long as the same type of material is simulated. Also, the dataset and experimental setups are identical across all models discussed in the paper. To conclude, **our DEL can effectively generalize to unobserved scenarios with different initial conditions** when simulating the same materials.
>
> Furthermore, in *Figure 8, 9 and Section G.3, G.4* in the Appendix, we evaluate the training efficiency of our model. The results show that even when trained with a limited amount of data, such as **1/8** training dataset, our model still achieves impressive performance. In contrast, other baselines that do not adopt physical priors exhibit a rapid decline in performance as the training data is reduced. This also indicates the generalization ability of our DEL.
>
> > **Weakness 3: The Large Number of Particles**
>
> Thanks for this comment. We indeed have not conducted experiments in particularly large scenes because representing large scenes with particles requires a substantial amount of VRAM. However, unlike other methods that model the dynamics across the entire spatial domain, our approach models only the interactions between particles, analyzing each pair individually. This way significantly **reduces the computational overhead** when simulating relatively large scenes, and is **scale-independent**. Due to its scale-invariant properties, our method should outperform others when simulating large scenes. In the future, we plan to explore hierarchical simulation techniques to reduce memory consumption, thereby improving the efficiency of simulating larger scenes.
>
> > **Question 1: GPF related**
>
> Thanks for this question. The GPF does not need to be retrained for different scenes, because GPF is a **generalizable point-based NeRF** method, it has already been pretrained on large 3D reconstruction datasets and can directly convert multiview images into a point-based NeRF representation without backpropagation. Moreover, we slightly fine-tuned the GPF on our training set based on their provided checkpoints for better reconstruction quality.
>
> To improve the readability of this paper, we introduce the above basic preliminaries about GPF in *the first paragraph of Section 3* following this comment. Moreover, we add detailed descriptions of the point-based renderer that we use in our revised Appendix.
>
> Besides, we also evaluate the performance of our method when different point-based renderers are adopted in *Section G2, Figures 10 and 11* in the Appendix. The results illustrate that DEL is robust to different renderers. As long as the renderer meets the following requirements: **1.** Point-based: can change the content with the movement of points. **2.** Differentiable: can propagate gradient, it can be used to train DEL.
>
> > **Question 2: Long-term prediction**
>
> The average timestamps for each scenario in the test set can be seen in the following Table:
>
> | Scenario | plasticine | SandFall | Multi-Obj | FluidR | Bear | Fluids |
> |-|-|-|-|-|-|-|
> |Avg Step Number | 86 | 94 | 132 | 155 | 124 | 138 |
>
> Among them, the Plasticine and SandFall predict relatively short-term dynamics, while **the Multi-Obj, FluidR, and Fluids provide the results for long-term predictions** whose timestamps are approximately **from 130 to 160**. In addition, all metrics reported in this paper including *CD, EMD, PSNR, SSIM, and LPIPS* are averaged across all timestamps of all sequences in the test set. This indicates that our method can perform better than these counterparts in both long-term and short-term predictions. We are also going to explore more super-long examples of these learned simulators in the follow-up plan.
>
> > **Question 3: A typo in the definition of loss function**
>
> We thank the reviewer's careful check very much. This is a typo in *Line 253*, the loss should be L1 loss and we have corrected it in our revised paper.

---

### Official Review · Reviewer_qBCD · 2024-07-12

**Soundness:** 3
**Presentation:** 3
**Contribution:** 3
**Rating:** 6
**Confidence:** 4

**Summary:**

The paper proposes a framework to learn 3D dynamics from 2D observations. DEM uses hand-designed kernel functions to model interaction force between particles, which may vary significantly for different material types. The paper changes these kernels to learnable GNN kernels. The time integration still follows DEM. Combined with a differentiable particle renderer, these kernels can be trained to match 2D observations.

**Strengths:**

The framework is quite general, with force-based time integration providing a strong physics prior. The use of learnable kernels to model point interactions enables the DEM to effectively fit observations.

Numerous comparisons are conducted, and the results are promising.

**Weaknesses:**

The evaluations are conducted only on synthetic data.

The learned particle interactions are black-boxes, which are not explainable.

**Questions:**

[Learning Neural Constitutive Laws From Motion Observations for Generalizable PDE Dynamics](https://arxiv.org/abs/2304.14369) should be cited, since it solves a similar task.

Are the material types in particle attribute A pre-known? Are the initial particle distributions known? The known assumptions should be listed in the paper.

How are particle colors assigned? In reality, objects usually have textures. Manually assigning colors may be impractical.

The particle initialization step is illustrated in Fig. 1, but no technical details are discussed in the text. It seems that the initial geometry is provided in the experiments. In that case, the known-geometry assumption should be illustrated in the figure instead. However, geometry reconstruction is important for real data applications.

**Limitations:**

Limitations are discussed.

---

> ### Author Rebuttal · Authors · 2024-08-03
>
> > **Weakness 1: The evaluations are conducted only on synthetic data.**
>
> Thanks for this comment. At present, the proposed method is indeed only evaluated on synthetic data. This is because capturing multiview videos for a dynamic process in the real world is difficult. Because it is hard to **simultaneously capture multiple dynamic videos** of the same object. Additionally, some objects, such as plastic materials, may become damaged via collision when shooting the first video. So it is **hard to be repeated**. We have also pointed out this issue in *the Limitation section* of our original paper.
>
> In the future, we are going to investigate potential solutions for that. First, transferring the model trained on synthetic data to real-world situations might be possible. Second, we are going to explore few-shot or one-shot learning to learn dynamics with a single real-world video, which might be achieved by integrating more explicit physics and geometry priors into learning-based frameworks.
>
> > **Weakness 2: The learned particle interactions are black-boxes, which are not explainable.**
>
> The DEL is proposed by **integrating graph neural operators into a classic mechanics analysis framework**, we intentionally make these operators to approximate the mapping from particle deformation to particle interaction forces. Therefore, we would like to claim that this approach is **partially interpretable**. We list the reasons as follows:
>
> **First**, we directly adopt the predicted particle interaction forces to update the velocities and positions of all particles, and achieve accurate results, which can indicate **the predicted forces are correct** since **incorrect force cannot result in precise deformations** under the framework of Euler integration.
>
> **Second**, since we employed the framework of the Discrete Element Method (DEM), it naturally satisfies **the conservation of momentum and energy**. In addition, given our assumption that particle mass remains constant and particles neither vanish nor are created spontaneously, our method also adheres to the law of **conservation of mass**.
>
> **Third**, we conduct additional analysis to evaluate the traits of these neural operators in *Section H and Figure 17* in the Appendix. We visually **study how the predicted forces relate to the particle deformation** of different materials. The relationships are implicitly learned by our DEL. We observe that the **force-deformation curves indeed reflect the real-world properties** of these materials.
> There are two examples to illustrate. **(1)** With plastic materials, the force grows as they stretch at the beginning. But after stretching to a certain extent, they begin to yield, and more displacement would not increase forces further. This behavior is truly like real plastic materials. **(2)** For rigid solid materials, even tiny shape changes can cause extremely large force to maintain their initial shapes, which also matches what we see in the real world.
>
> > **Question 1: Cite a relevant paper**
>
> We appreciate this reviewer for bringing such a relevant paper to us. We cite the paper in a proper place in our revised paper.
>
> > **Question 2 and 4: Known Assumptions and Particle Initialization**
>
> We would like to present that both the material types and initial particle distribution are also **unknown** in our DEL framework. In *particle attribute A*, we only tell the DEL which particles belong to the same material, but we **do not provide the certain material type** for them. Only the image sequences and their corresponding camera poses are needed to train the model. The material types, including their mechanical properties, can be **implicitly encoded into the graph neural operators** during training.
>
> The initial particle distribution can be obtained by using the Generalizable Point Field (**GPF**) [1] in this work, which is a differentiable PointNeRF-like 3D reconstruction method. We introduce this step in *Section 3.1*. We would like to claim that our method can be seamlessly combined with other point-based renderers, as long as they can **(1)** render the deformed scene according to the point movement and **(2)** propagate the gradients. The simulation starts only after the initialization of the particles is finished. Moreover, we evaluate our method on the other two different differentiable point-based renderers, i.e. ParticleNeRF and PhysNeRF in Section G2 and Figures 10 and 11 in the Appendix, the results show that our DEL can also **deliver the best performance no matter what renderers are used**.
>
> Following the reviewer's suggestion, we add detailed descriptions of the **assumption and initialization** of particles in *the first paragraph of Section 3* in our revised paper. Furthermore, to improve the readability, we add some basic background knowledge of the GPF in our revised Appendix.
>
> *[1] Wang J, et al.. Learning robust generalizable radiance field with visibility and feature augmented point representation. International Conference of Learning Representations (ICLR) 2024*
>
> > **Question 3: Color and Texture**
>
> As we stated in the answer to Question 1, the colors of these particles are faithfully reconstructed from images by using GPF. These colors and textures would remain unchanged during the simulation process. We agree with the review that the textures in reality could be various and complicated. However, we would argue that **complex textures are more beneficial for learning dynamics because they better reveal the direction of deformation at specific points on an object's surface due to the color gradients in the image**. This helps in reducing uncertainty. However the uniform color does not have large gradients. Conversely, textures of an object with uniform color make it difficult to ascertain the exact direction of deformation at a point on the surface, because there are no color gradients around that point.

---

> > ### Comment · Reviewer_qBCD · 2024-08-14
> >
> > Thank you for the rebuttal. I do not have further questions.

---

> > > ### Author Response · Authors · 2024-08-14
> > >
> > > We sincerely thank the reviewer's constructive suggestions, time, and attention in the review and discussion periods, which help us a lot in polishing our manuscript and updating the revision version.

---

### Official Review · Reviewer_LfJV · 2024-07-13

**Soundness:** 4
**Presentation:** 2
**Contribution:** 4
**Rating:** 6
**Confidence:** 3

**Summary:**

The paper considers the problem of physical modeling the dynamics of 3D objects in space using only 2D observations. The authors propose to solve this problem by viewing objects as sets of interacting points and using differentiable rendering of these point clouds. In this pipeline, neural models are used to directly predict the dynamics of 3D particles. Unlike existing methods, the authors propose to use a physically principled framework (Discrete Element Analysis) to develop a partially interpretable neural model that can be used to predict the target dynamics. This framework decomposes the forces applied to the points into gravity, potential, and viscous interaction forces (and further into normal and tangent components). The proposed model thus is constrained to predict certain components from these force decompositions resulting in a more principled, interpretable, and effective dynamics prediction.

The authors propose new more challenging synthetic datasets containing variable materials and objects, evaluate on this data, and compare the results to several existing baselines. The proposed method shows improvements over the baselines in all the scenarios. The authors also provide an ablation study showing the importance of different components.

**Strengths:**

* The proposed model seems like an effective combination of a physically principled framework and neural model, which potentially can be further explored and improved.
* The proposed method significantly outperforms all baselines.
* The authors propose a new dataset.

**Weaknesses:**

* The written text requires some additional polishing (citations should not be treated as nouns, sometimes the layout of figures and text is too tight, and there are some grammatical errors and not correctly formulated sentences).

**Questions:**

How are CD and EMD computed? Are any of the intermediate states used to compute the metrics or are they only computed for the final state? If the proposed model is interpretable does it make sense to compare the predicted particle forces in all the intermediate steps with the ground truth forces from the simulations?

Some examples show that objects can be «torn» over the course of the considered scenario. Currently, the resulting distinct parts of the torn object will still be considered as one single object by the model, is not it? Does it cause problems? Have you considered any solutions for it?

**Limitations:**

The limitations are properly addressed in the text.

---

> ### Author Rebuttal · Authors · 2024-08-03
>
> > **Weakness: Polishing Language**
>
> We thank the reviewer for pointing out this. Following this comment, we have thoroughly revised the language and grammar to enhance the overall quality of the text, carefully checked the citation, and optimized the layout. The updated version is fully improved for the academic language.
>
> > **Question 1: Metrics Calculation**
>
> As stated by the reviewer, our goal is to predict a sequence of particle trajectories, therefore there are lots of intermediate states. We would claim that **all the metrics** in this work, including *CD, EMD, PSNR, SSIM, and LPIPS*, are derived by averaging over all timestamps of all sequences in the test set. Hence the evaluated metrics contain **all intermediate states and all sequences**. We will clarify this point in *Section 4* of our revised manuscript following this comment.
>
> > **Question 2: Interpretable**
>
> Thanks for this comment. Yes, the proposed method is partially interpretable because we predict the interaction forces between particles by our physical neural operators. The predicted force is directly applied to update the velocity and position of particles by Euler Integration. Therefore, the next accurate positions can **only be obtained by correct force prediction**. However, we do not have the groundtruth of the particle interaction forces. The reason is that all the synthetic data are generated by the Material Point Method (MPM). In MPM, the interaction forces are not explicitly and directly derived. Instead, it simulates the dynamics by transferring the momentum between particles and background grids. Therefore, it's hard to obtain the groundtruth of the interaction forces.
>
> Nevertheless, we conducted additional analysis to validate **the physical meaning of the forces** predicted by our DEL. In *Section H and Figure 17 in the Appendix*, we visualize the **relationship between the predicted forces and particle deformations** for each type of material by accessing the learned graph operators with different inputs. These curves indeed reveal the inherent mechanical properties of these different materials. For example, for plastic materials, the force initially increases with displacement. However, when the deformation reaches a certain extent, the material undergoes yielding, and the force no longer increases with further displacement or even decreases. This phenomenon closely mirrors real-world plastic materials. Moreover, for the rigid body, even very small deformations can lead to a significant increase in force, which is also in agreement with real-world situations.
>
> > **Question 3: Torning Objects**
>
> In fact, the proposed method recognizes particles in terms of the **material level** instead of the object level. If an object is torn apart, **the material of these parts remains unchanged**, therefore they have the same mechanical properties. For example, a piece of plastic, even when torn into two halves, remains plastic. This phenomenon looks like the DEL recognizes them as the same objects, but as we stated above, these particles just belong to the same material with constant properties, such as mechanical parameters, colors, etc. Hence DEL does not meet these object-level issues.

---

### Decision · Program_Chairs · 2024-09-25

**Decision:**

Accept (poster)

**Comment:**

The reviewers are generally positive on the paper.  Some questions were raised around generalization, evaluation, and scaling and some minor presentation issues were presented.  Overall, these questions have been resolved in the rebuttal and discussion phase and the presentation issues are minor and able to be resolved in the final version.

As such, the AC recommends acceptance as a poster and encourages the authors to carefully revise the paper based on the feedback from the review process.